# TGFβ signaling directs serrated adenomas to the mesenchymal colorectal cancer subtype

Evelyn Fessler[1,2], Jarno Drost[2,3], Sander R van Hooff[1,2], Janneke F Linnekamp[1,2], Xin Wang[4], Marnix Jansen[5,†], Felipe De Sousa E Melo[1,2,‡], Pramudita R Prasetyanti[1,2], Joep EG IJspeert[6], Marek Franitza[7,8], Peter Nürnberg[7,8], Carel JM van Noesel[5], Evelien Dekker[6], Louis Vermeulen[1], Hans Clevers[2,3] & Jan Paul Medema[1,2,*]

## Abstract

The heterogeneous nature of colorectal cancer (CRC) complicates prognosis and is suggested to be a determining factor in the efficacy of adjuvant therapy for individual patients. Based on gene expression profiling, CRC is currently classified into four consensus molecular subtypes (CMSs), characterized by specific biological programs, thus suggesting the existence of unifying developmental drivers for each CMS. Using human organoid cultures, we investigated the role of such developmental drivers at the premalignant stage of distinct CRC subtypes and found that TGFβ plays an important role in the development of the mesenchymal CMS4, which is of special interest due to its association with dismal prognosis. We show that in tubular adenomas (TAs), which progress to classical CRCs, the dominating response to TGFβ is death by apoptosis. By contrast, induction of a mesenchymal phenotype upon TGFβ treatment prevails in a genetically engineered organoid culture carrying a *BRAF*[V600E] mutation, constituting a model system for sessile serrated adenomas (SSAs). Our data indicate that TGFβ signaling is already active in SSA precursor lesions and that TGFβ is a critical cue for directing SSAs to the mesenchymal, poor-prognosis CMS4 of CRC.

**Keywords** cancer subtypes; colorectal cancer; epithelial–mesenchymal transition; sessile serrated adenoma; transforming growth factor beta (TGFβ)
**Subject Categories** Cancer; Digestive System

## Introduction

Transforming growth factor-β (TGFβ) signaling controls a plethora of physiological programs, influencing cellular behavior and tissue homeostasis. The response to TGFβ is highly context specific and results in distinct or even opposite effects for different cell types (Massague, 2012). The multifunctional nature of this cytokine can also be observed in tumorigenesis, where it plays either a tumor-suppressing or tumor-promoting role (Massague, 2008). Induction of apoptosis, for example, by upregulation of the pro-apoptotic molecule BIM and the death-associated protein kinase (DAPK) (Jang *et al*, 2002; Ramesh *et al*, 2008; Heldin *et al*, 2009), as well as the arrest of proliferation by the induction of cyclin-dependent kinase inhibitors such as p21[CIP1], are examples of tumor-suppressive mechanisms (Hannon & Beach, 1994; Datto *et al*, 1995; Massague, 2008). On the other hand, tumors can benefit from activated TGFβ signaling due to its ability to enhance cancer cell invasion and metastasis. The TGFβ signaling molecule is a well-known inducer of the epithelial–mesenchymal transition (EMT) (Moustakas & Heldin, 2007), a process during which epithelial-organized cells lose their cell–cell junctions and gain a mesenchymal, migratory, and invasive phenotype, allowing them to leave the primary site and spread to distant organs (Thiery, 2002). Recent insight from murine models, however, proposes that EMT may be related to therapy resistance rather than enhanced metastatic spread (Fischer *et al*, 2015; Zheng *et al*, 2015). Regardless of the exact mechanism, the mesenchymal phenotype has been linked to dismal prognosis in patients for many cancer types, including colorectal cancer (CRC) (Guinney *et al*, 2015). CRC samples can be classified into four distinct consensus molecular subtypes (CMSs), of which CMS4 displays a

1 Laboratory for Experimental Oncology and Radiobiology (LEXOR), Center for Experimental Molecular Medicine (CEMM), Academic Medical Center (AMC), University of Amsterdam, Amsterdam, The Netherlands
2 Cancer Genomics Center, Amsterdam, The Netherlands
3 Hubrecht Institute, Royal Netherlands Academy of Arts and Sciences (KNAW) and UMC Utrecht, Utrecht, The Netherlands
4 Department of Biomedical Sciences, City University of Hong Kong, Kowloon Tong, Hong Kong
5 Department of Pathology, AMC, University of Amsterdam, Amsterdam, The Netherlands
6 Department of Gastroenterology, AMC, University of Amsterdam, Amsterdam, The Netherlands
7 Cologne Center for Genomics (CCG), University of Cologne, Cologne, Germany
8 Cologne Excellence Cluster on Cellular Stress Responses in Aging-Associated Diseases (CECAD), University of Cologne, Cologne, Germany
*Corresponding author. Tel: +31 20 56 67777; E-mail: j.p.medema@amc.nl
†Present address: Centre for Tumour Biology, Barts Cancer Institute, University of London, London, UK
‡Present address: Department of Molecular Oncology, Genentech Inc., San Francisco, CA, USA

mesenchymal gene expression profile and dismal clinical outcome (Guinney *et al*, 2015).

Even though components of the TGFβ signaling pathway are often inactivated during CRC progression (Markowitz *et al*, 1995; Fleming *et al*, 2013), it has been shown that active TGFβ signaling—judged by phosphorylation of the downstream components SMAD2/3 (phospho-SMAD2/3)—can be found in colorectal carcinomas (Brunen *et al*, 2013). Tumors associated with poor prognosis display significantly higher levels of phospho-SMAD2/3 also in the tumor epithelium and activation of the TGFβ pathway has been linked to resistance to conventional and targeted chemotherapeutics (Huang *et al*, 2012; Brunen *et al*, 2013).

For a long time, the development of CRC was thought to follow a molecularly well-defined route with the inactivation of the adenomatous polyposis coli (*APC*) gene as the initiating event followed by the activation of the *KRAS* oncogene and the inactivation of *TP53* as the tumor progresses to a metastatic carcinoma (Fearon & Vogelstein, 1990). The tubular adenoma (TA) was viewed as the main epithelial precursor lesion to spawn colorectal malignancies (Muto *et al*, 1975). However, over the last two decades, it has become increasingly clear that the heterogeneity observed in colorectal malignancies is also reflected already at the premalignant stage (IJspeert *et al*, 2015). The classical path of CRC development and progression has been complemented with the serrated neoplasia pathway (Snover, 2011). Several types of serrated polyps have been described (IJspeert *et al*, 2015), and the sessile serrated adenoma (SSA) could be linked to malignant progression to CRC (Oono *et al*, 2009; Lash *et al*, 2010; IJspeert *et al*, 2015). Histologically and molecularly SSAs present as distinct entities, characterized by a serrated morphology and the activation of the *BRAF* oncogene (Leggett & Whitehall, 2010; Snover, 2011). Furthermore, serrated lesions often display DNA hypermethylation of CpG islands in promoter regions, leading to silencing of tumor suppressor genes (also known as the CpG island methylator phenotype or CIMP) (Park *et al*, 2003; Kambara *et al*, 2004). Specific precursor lesions have been suggested to develop into different types of CRC based on gene expression profiling and SSAs were suggested to harbor the potential to develop into the mesenchymal, poor-prognosis CRC subtype, whereas TAs more closely relate to the chromosomally instable type of CRC (De Sousa E Melo *et al*, 2013). However, it is unclear what is responsible for promoting subtype-specific transformation and for installing unique features associated with distinct groups of CRC. Gene expression-based characterization of a small set of TA and SSA samples has allowed a glimpse of the wiring of these polyps. Whereas components of the WNT pathway are highly expressed in TAs, SSAs present with high levels of EMT- and TGFβ pathway-associated genes (De Sousa E Melo *et al*, 2013). Therefore, we set out to determine the response of genetically distinct CRC precursor lesions to TGFβ stimulation in organoid cultures, model systems that closely recapitulate human disease (van de Wetering *et al*, 2015). We made use of human organoid cultures from normal colon tissue and TA polyps, and the CRISPR (clustered regularly interspaced short palindromic repeat)-Cas9 (CRISPR-associated nuclease 9) system to genetically engineer an organoid culture to carry the *BRAF*^V600E mutation, thus modeling different paths of CRC development. In TA organoids, which most frequently progress to the classical CRC subtype, apoptosis was the dominating response upon TGFβ treatment, while induction of a mesenchymal phenotype

prevailed in the *BRAF*^V600E-mutated organoid culture, presenting a model system for the serrated path to CRC. SSAs following the serrated neoplasia pathway have been suggested to harbor the potential to progress either to good- or to poor-prognosis CRCs based on molecular markers (Jass, 2007; Phipps *et al*, 2015). Indeed, using gene expression data, we show that SSAs can be segregated into both CMS1- and CMS4-like lesions, which is associated with activation of the TGFβ pathway. Significantly higher levels of TGFβ pathway activity were detectable in SSAs predicted to progress to CMS4-like CRCs compared to those poised to give rise to CMS1 CRC. Hence, our data point to an important role of TGFβ in the serrated path of CRC development and propose that this cytokine represents a critical cue in directing SSAs to the mesenchymal, poor-prognosis CRC subtype.

# Results

## TGFβ induces apoptosis in human tubular adenoma organoid cultures

To study the effect of TGFβ at an early stage of tumor development, we obtained TAs from familial adenomatous polyposis (FAP) patients, which would predictably follow the classical path of CRC development (Fearon & Vogelstein, 1990; De Sousa E Melo *et al*, 2013). Organoid cultures (TA1–TA5) were established from these premalignant lesions and they were propagated in medium without the WNT-ligands WNT3A and R-Spondin-1 to select for transformed cells in which the WNT pathway is constitutively active. Normal non-transformed cells without activated WNT signaling do not survive this selection process (Drost *et al*, 2015). Of note, besides WNT pathway activation, one of the TA organoid cultures—TA1—also carries a *KRAS*^G12V mutation (Appendix Fig S1A). In the classical path of CRC development, activating mutations in the *KRAS* oncogene are thought to follow aberrant WNT pathway activation (Fearon & Vogelstein, 1990); hence, the origin of the TA1 organoid culture is likely to be a more advanced adenoma or early carcinoma. The TGFβ pathway was not perturbed in the TA organoid cultures, as all five cultures used in this study showed SMAD4 expression and induction of phopho-SMAD2 upon TGFβ stimulation (Appendix Fig S1B and C). In the organoid cultures TA2-TA5, the formation of well-organized structures was disrupted by the addition of TGFβ to the culture medium, leading to disintegration of the organoids (Fig 1A). In the control condition, cleaved Caspase-3-positive, and thus apoptotic, cells could only be found inside the organoids (Fig 1B). These represent old cells that have been replaced by a new generation, undergo apoptosis, and are shed into the lumen. In contrast, the amount of cleaved Caspase-3-positive cells was strongly increased upon TGFβ treatment, highlighting the loss of organization in these structures (Fig 1B). In contrast, the *KRAS*-mutant TA1 organoid culture did not undergo apoptosis judged by the lack of cleaved Caspase-3, but rather showed growth arrest both morphologically and based on KI-67 expression (Figs 1A and B, and EV1A). We performed gene expression arrays on control and TGFβ-treated samples of three TA organoid cultures and two organoid cultures from normal colon tissue. As expected, genes involved in apoptosis were highly enriched in the TGFβ-treated compared to the control samples judged by gene set enrichment

    

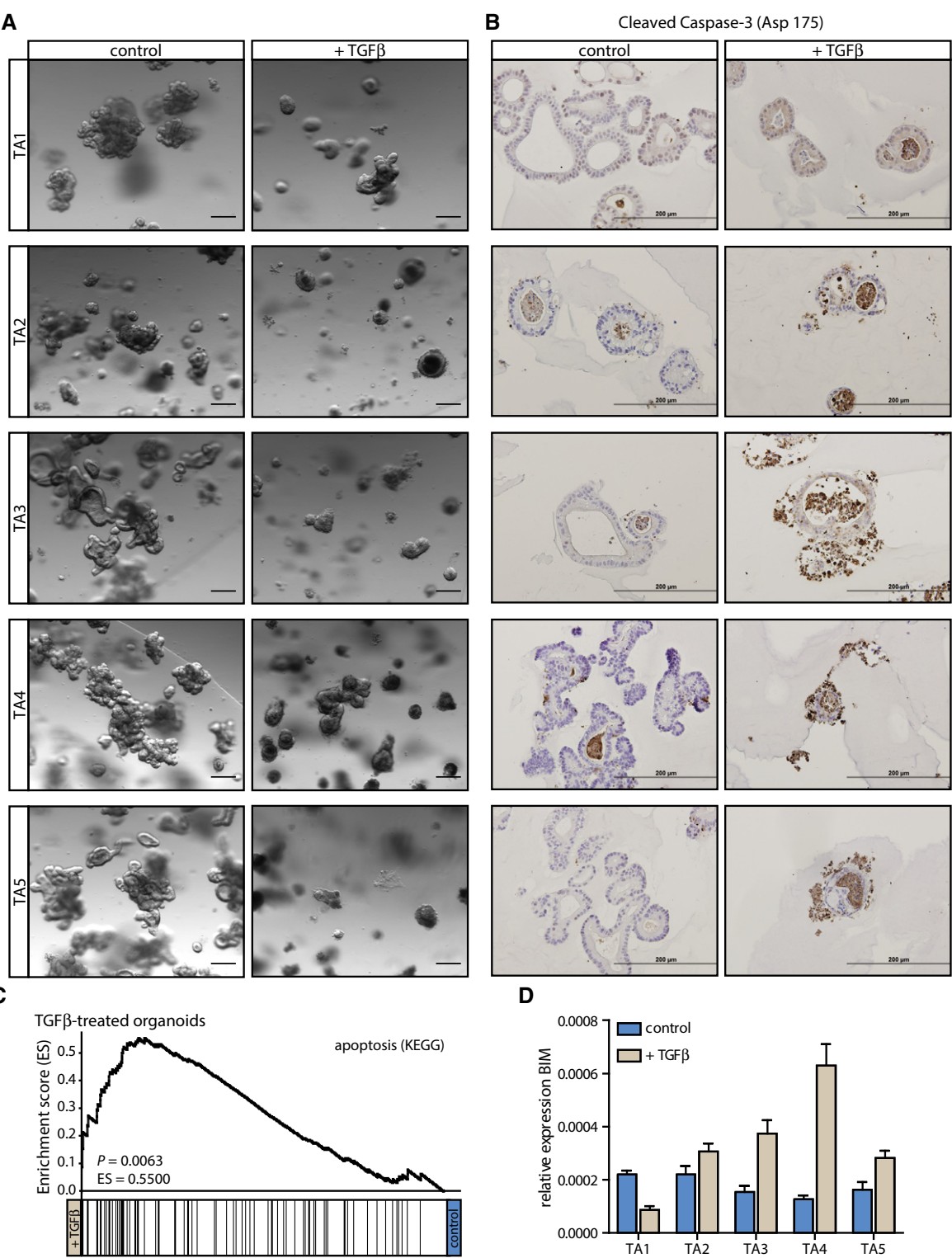

**Figure 1.  TGFβ induces an apoptotic response in human tubular adenoma (TA) organoids.**

A   The usually well-organized human TA organoids disintegrate upon TGFβ stimulation (scale bars: 200 μm).

B   Following TGFβ treatment, TA organoid cultures display increased levels of cleaved Caspase-3 (scale bars: 200 μm).

C   Gene expression profiles of TGFβ-treated organoids show enrichment in apoptosis-related genes when compared to the control condition.

D   The pro-apoptotic molecule BIM (*BCL2L11*) is upregulated in the TGFβ-treated condition in all TA cultures, besides in TA1, which carries a *KRAS*$^{G12V}$ mutation (one representative of ≥ 3 independent experiments is shown, error bars represent SD).

Source data are available online for this figure.

analysis (GSEA) (Mootha *et al*, 2003; Subramanian *et al*, 2005) (Fig 1C). Similar results have previously been reported for mouse intestinal organoids carrying an inactivating *Apc* mutation, in which BCL2-like protein 11 (*BCL2L11*, BIM) was identified to mediate TGFβ-induced apoptosis (Wiener *et al*, 2014). Also in the human TA organoid cultures, BIM was induced upon TGFβ treatment (Fig 1D), whereas BID and Puma (*BBC3*) were not induced or down-regulated (Fig EV1A). All TA cultures displayed upregulation of BIM, except the *KRAS*-mutated TA1 organoid culture (Fig 1D), con-firming the previously published results that addition of a *KRAS* mutation to an *APC*-mutated background increased resistance to TGFβ-mediated apoptosis by inhibiting the induction of BIM (Wiener *et al*, 2014). Also human wild-type organoid cultures isolated from normal colon mucosa did not show induction of BIM and cleaved Caspase-3 upon TGFβ treatment (Fig EV1B and C), but slowed proliferation illustrated by reduction of KI-67 expression (Fig EV1C). These results indicate that human cells with activated WNT signaling respond to TGFβ via induction of apoptosis, which is alleviated in the presence of an oncogenic *KRAS* mutation. Thus, patient-derived organoid cultures from premalignant lesions recapit-ulate similar phenotypes upon TGFβ stimulation as organoids from genetically engineered mouse models (Wiener *et al*, 2014).

## Human colon organoid cultures respond to TGFβ by induction of EMT features

TGFβ is a well-known inducer of the EMT program (Moustakas & Heldin, 2007). Indeed, we observed morphological changes in the normal colon organoid cultures that resembled the induction of a mesenchymal phenotype in these usually well-organized epithelial structures (Fig 2A). Also in all five TA organoid cultures morpho-logical changes indicative of EMT induction could be observed in the surviving cells (Fig 2B and Appendix Fig S2A). Applying EMT signatures to the gene expression data obtained from TGFβ-treated and control organoids revealed that genes present in these signa-tures were enriched in TGFβ-treated compared to control samples (Fig 2C and Appendix Fig S2B) (Taube *et al*, 2010; Gröger *et al*, 2012). In accordance, the EMT-inducing transcription factor zinc finger E-box binding homeobox 1 (ZEB1) was upregulated upon TGFβ treatment in both normal colon and TA organoid cultures (Fig 2D and E). Also the mesenchymal marker fibronectin 1 (FN1) was strongly induced upon TGFβ treatment (Fig 2F and Appendix Fig S2C). Taken together, non-transformed as well as transformed colon organoid cultures possess the ability to respond to TGFβ via the induction of the EMT program.

## TGFβ-treated human organoids adapt a mesenchymal CRC subtype gene expression profile

Recently, CRC has been classified into multiple subtypes by several groups (Cancer Genome Atlas Network, 2012; Perez-Villamil *et al*, 2012; Schlicker *et al*, 2012; Budinska *et al*, 2013; De Sousa E Melo *et al*, 2013; Marisa *et al*, 2013; Sadanandam *et al*, 2013; Roepman *et al*, 2014), and in a large international effort, the CRC subtyping consortium (CRCSC) unified these classifications resulting in the four consensus molecular subtypes (CMSs) of CRC (Guinney *et al*, 2015). In line with the individual classifications, a mesenchymal subtype (CMS4) was identified with significantly worse recurrence-free

survival (RFS) (Guinney *et al*, 2015). This subtype was characterized by high expression of EMT-associated genes (De Sousa E Melo *et al*, 2013; Guinney *et al*, 2015). Additionally, based on gene expression profiling SSAs were predicted to be potential precursor lesions of this mesenchymal, poor-prognosis colon cancer subgroup (De Sousa E Melo *et al*, 2013). Since the EMT phenotype—a hallmark of the mesenchymal colon cancer subtype—can be induced by TGFβ, and the TGFβ signaling pathway is predicted to be active based on gene expression in CMS4 CRCs (Guinney *et al*, 2015) and SSAs (De Sousa E Melo *et al*, 2013), we wondered whether TGFβ might also dictate subtype-specific gene expression. We chose FRMD6 (FERM domain containing 6), a marker that is highly expressed in tumors of the mesenchymal subtype, and caudal-type homeobox 2 (CDX2), which is highly expressed in tumors of the classical group (De Sousa E Melo *et al*, 2013) and assessed their expression changes upon TGFβ treat-ment. Normal colon and TA organoid cultures responded to TGFβ treatment with upregulation of FRMD6 (Fig 3A and B). CDX2 levels were strongly reduced upon TGFβ stimulation both on RNA and protein level (Fig 3A–C and Appendix Fig S2D). A direct comparison of the gene expression changes induced by TGFβ in organoids with either genes upregulated in CMS4 cancers (Fig 3D) or the 500 most highly expressed genes in SSAs compared with TAs (Fig 3E) con-firmed this switch to a more CMS4/SSA-like profile (Fig 3D and E). Therefore, TGFβ treatment of colon organoids not only induced EMT, but also induced the expression of CMS4/mesenchymal marker genes and downregulated expression of genes associated with the classical, epithelial type of CRC.

## SSA and TA polyps can be separated using a TGFβ signature, which is predictive of prognosis in CRC datasets

To gain further insight into the role of TGFβ signaling at the early stages of tumor development *in vivo*, we generated gene expression data from a set of TA and SSA polyps (GSE45270 and GSE79460) and analyzed the expression of the five most strongly induced and five most strongly reduced genes upon TGFβ treatment of TA orga-noid cultures. The expression of these genes followed the expected pattern, meaning genes induced upon TGFβ treatment were more highly expressed in SSAs, while the downregulated genes could be detected at higher levels in TAs (Fig 4A). Applying an *in vitro* TGFβ response signature from TGFβ-treated organoid cultures to gene expression data from SSAs and TAs revealed that this signature was able to segregate SSA from TA samples (Fig 4B). Additionally, this epithelial cell-derived TGFβ signature was capable of clustering most of the CMS4 samples of the AMC-AJCCII-90 dataset apart from CMS1-3 tumor samples (Fig EV2B) (GSE33113, De Sousa E Melo *et al*, 2011). GSEA confirmed the higher expression of the TGFβ signature genes in SSA lesions and CMS4 tumors (Fig EV2A). As the TGFβ signature is able to identify CMS4 tumors that reportedly display worse RFS (Guinney *et al*, 2015), we speculated that this signature would be able to predict RFS. Indeed, our *in vitro* TGFβ response signature could predict RFS in two independent CRC datasets (Fig EV2C). Importantly, the gene expression-based obser-vations were substantiated by the finding that the TGFβ target gene ZEB1 was strongly expressed in the epithelium of SSA but not TA polyps at the protein level (Fig EV3A).

Based on molecular markers, such as *BRAF* mutation and CIMP, SSAs have previously been suggested to develop into two distinct

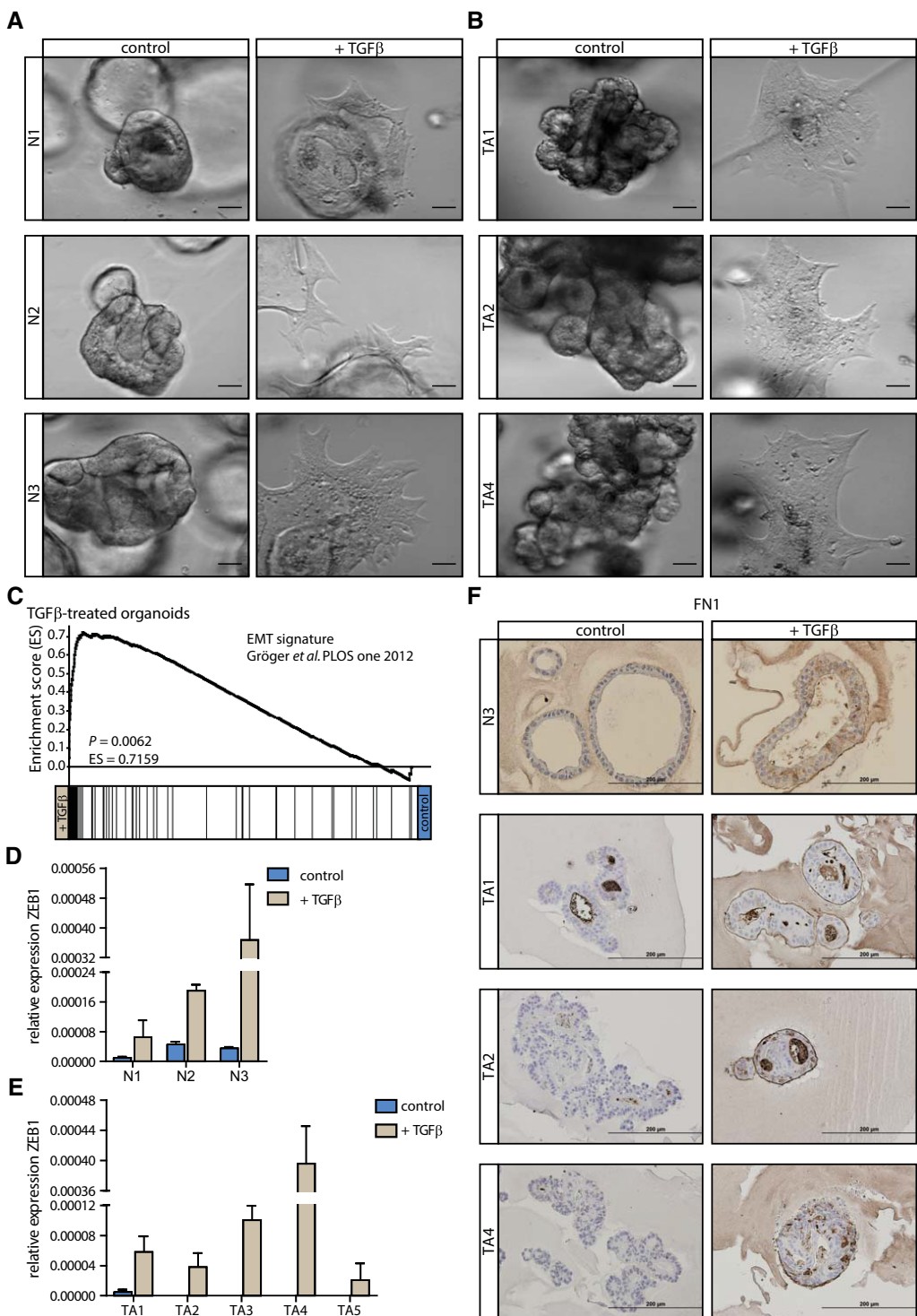

**Figure 2.  TGFβ stimulation of human colon organoids induces EMT features.**

A, B    Morphological changes in (A) normal colon (N1-3) and (B) surviving TA organoids upon TGFβ treatment resemble the induction of a mesenchymal phenotype in these usually well-organized epithelial structures (scale bars: 50 μm).

C    Genes present in an EMT signature (Gröger *et al*, 2012) are enriched in TGFβ-treated compared to control samples.

D, E    The EMT-inducing transcription factor ZEB1 is strongly induced upon TGFβ treatment both in (D) normal [one (representative) experiment is shown (*n* = 1 for N1 and N2, and *n* = 3 for N3), error bars represent SD] as well as (E) TA  (one representative of ≥ 3 independent experiments is shown, error bars represent SD) organoids.

F    The mesenchymal marker fibronectin 1 (FN1) is induced following TGFβ stimulation (scale bars: 200 μm).

Source data are available online for this figure.

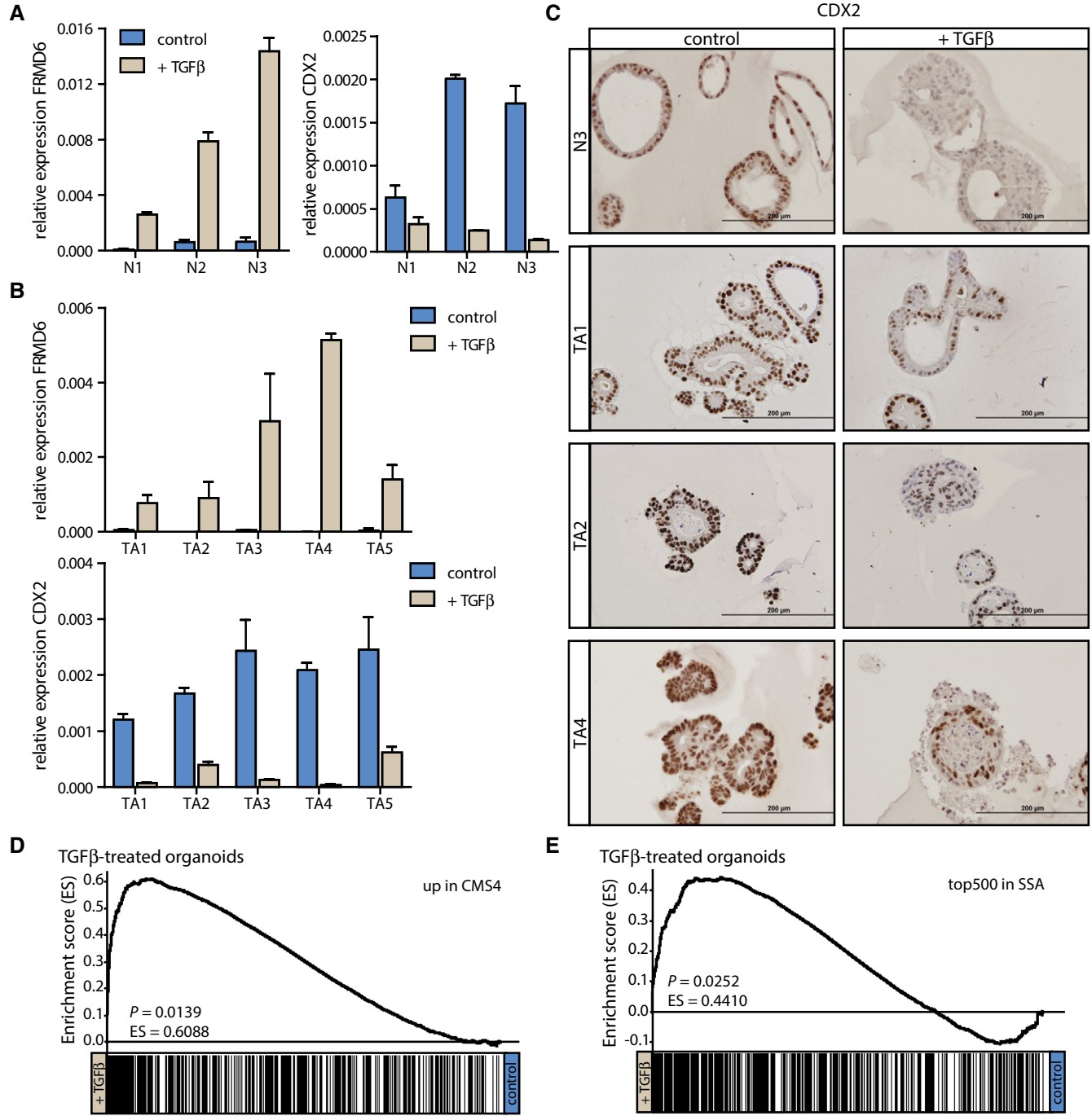

**Figure 3. TA organoids adapt their gene expression profiles to those of CMS4/SSA samples upon TGFβ treatment.**

A, B  FRMD6 expression is strongly induced and CDX2 expression is downregulated both in (A) normal [one (representative) experiment is shown ($n = 1$ for N1 and N2, and $n = 3$ for N3), error bars represent SD] and (B) TA (one representative of $\geq 3$ independent experiments is shown, error bars represent SD) organoid cultures following TGFβ stimulation.

C  The reduction in CDX2 levels in the TGFβ-treated condition is also observed on the protein level (scale bars: 200 μm).

D, E  Gene expression profiles of TGFβ-treated samples show enrichment of genes highly expressed in (D) the mesenchymal CMS4 of CRC (log2FC > 2) and (E) SSA precursor lesions when compared to control samples.

Source data are available online for this figure.

types of CRC, one associated with good and the other one with poor prognosis (Jass, 2007; Phipps *et al*, 2015). Strikingly, we also observed that the *in vitro* TGFβ response signature derived from

organoids not only segregated TAs from SSAs, but also clustered the SSA samples into two distinct groups (Fig 4B). This indicated that one group of SSAs displays a more vigorous TGFβ response

as compared to the rest. To further study this distinction within the SSA population, the CRC subtype affiliation of these lesions was elucidated. CMS classification of the pre-neoplastic polyps allowed us to predict that SSAs could indeed progress to either the good-prognosis CMS1 (further referred to as CMS1-SSAs) or the poor-prognosis CMS4 (Fig EV3B) and this perfectly aligned with the segregation observed using the TGFβ signature. This classification also confirmed the link between TAs and the classical CMS2, as almost all TA samples were classified into the CMS2 group (Fig EV3B; see Fig EV3C for *KRAS* and *BRAF* mutation as well as CIMP status of these polyps). Intriguingly, the 4 SSAs that predictably would give rise to carcinomas of the CMS4 (further referred to as CMS4-SSAs) showed high expression of a subset of genes present in the *in vitro* TGFβ response signature (Fig 4B). Indeed, the genes most strongly induced upon TGFβ treatment and differentially expressed between TAs and SSAs showed significantly higher expression levels in CMS4-SSAs compared with CMS1-SSAs (Fig 4C). Taken together, these data indicate that (i) the TGFβ pathway is operational in premalignant SSAs, that (ii) SSAs are characterized by higher activity of TGFβ signaling compared to TAs, and that (iii) in CMS4-SSAs TGFβ activity is elevated even further, suggesting that high levels of TGFβ signaling direct SSAs to the CMS4 of CRC (Fig 4D).

## A genetically engineered human organoid culture as model for the serrated path to CRC

To further dissect the role of TGFβ signaling in SSA precursor lesions, we made use of a genetically engineered *BRAF*^V600E^-mutated organoid culture. To date, we and others have not been successful in establishing organoid cultures of SSAs (IJspeert *et al*, 2015), thus rendering direct analysis of the role of TGFβ signaling in these precursor lesions *in vitro* impossible. The serrated path of CRC development is thought to be initiated by a *BRAF*^V600E^ mutation (Snover, 2011). We therefore made use of the CRISPR (clustered regularly interspaced short palindromic repeat)-Cas9 (CRISPR-associated nuclease 9) system to genetically engineer an organoid culture to carry a *BRAF*^V600E^ mutation (Cho *et al*, 2013; Cong *et al*, 2013; Mali *et al*, 2013; Drost *et al*, 2015), thus modeling the serrated path to CRC. Importantly, this organoid culture does not display features of CIMP, yet, suggesting that it represents the earliest phase of the serrated pathway. The organoid culture carrying the *BRAF*^V600E^ mutation did not respond to TGFβ with apoptosis, judged by the lack of cleaved Caspase-3 and BIM induction (Fig 5A–C). As shown for normal and TA organoid cultures, the expression of the

pro-apoptotic genes BID and Puma was not induced upon TGFβ treatment (Appendix Fig S3A). Instead, the *BRAF*^V600E^-mutant organoid culture showed decreased expression of KI-67, indicative of growth arrest (Appendix Fig S3B). Additionally, TGFβ treatment induced a pronounced EMT-like phenotype in the *BRAF*^V600E^-mutant organoid culture, coinciding with a marked upregulation of ZEB1 and FN1 (Fig 5D and E). Furthermore, the organoid culture carrying the *BRAF*^V600E^ mutation strongly induced FRMD6 and reduced the expression of CDX2 (Fig 5F and G). These data indicate that lesions carrying distinct genetic insults respond differentially to TGFβ stimulation early in tumor development: Whereas an apoptotic response dominated in classical TAs, in both TA and *BRAF*^V600E^-mutated organoids a robust EMT response was observed upon TGFβ treatment. In the case of TA organoids, a minority of cells that was able to resist TGFβ-induced cell death was selected for, survived this stimulus, and underwent EMT. In contrast, the bulk of cells in the *BRAF*^V600E^-mutant organoid culture survived treatment with TGFβ and the majority were able to induce the EMT process (Fig EV4). By avoiding cell death upon stimulation with TGFβ *BRAF*^V600E^-mutant SSAs escape the detrimental effects of TGFβ signaling and could thus exploit TGFβ pathway activation by the induction of the EMT program. We therefore conclude that *BRAF*^V600E^ mutations in combination with a microenvironmental TGFβ signal could be the underlying pathway toward the induction of SSAs.

To further study the role of differential TGFβ signaling between CMS1-SSAs and CMS4-SSAs, gene expression profiles of the *BRAF*^V600E^-mutant organoid culture in the absence and presence of TGFβ were generated (GSE79461). The TGFβ response in the genetically engineered *BRAF*^V600E^-mutant organoid culture overlapped significantly with that of TGFβ-treated TA organoid cultures (Fig EV5A), indicating that TGFβ induces a similar cellular program in our engineered model system as in organoids from precursor lesions arisen in patients. In accordance with the previous observation using the TA TGFβ signature, genes induced upon TGFβ treatment in the *BRAF*^V600E^-mutant organoid culture were highly expressed in CMS4-SSAs compared to CMS1-SSAs and the most strongly downregulated genes followed the opposite pattern (Figs 5H and EV5B). Importantly, this distinction was maintained at the carcinoma stage where *BRAF*-mutant CMS4 tumors showed higher levels of TGFβ-induced target genes compared to *BRAF*-mutated CMS1 tumors and vice versa genes reduced upon TGFβ treatment displayed lower expression in *BRAF*-mutant CMS4 CRCs compared to CMS1 CRCs (Figs 5I and EV5C). Taken together, these data point to the fact that TGFβ is indeed a critical cue to direct SSA precursor lesions to the mesenchymal, poor-prognosis CMS4 of CRC

▶

**Figure 4.  TGFβ target genes are differentially expressed between TA and SSA samples.**

A  The five most strongly induced genes (right column) upon TGFβ treatment of TA organoid cultures are higher expressed in SSAs when compared to TAs, whereas the five most strongly reduced genes (left column) follow the opposite pattern (*n* = 12 for SSA and *n* = 15 for TA. The horizontal lines represent the mean, and the error bars display the SD. *P*-values are based on unpaired two-tailed Student's *t*-tests).

B  Hierarchical clustering of 15 TA and 12 SSA samples using a TGFβ response signature derived from TGFβ-treated TA organoids (|log2FC| > 3). Expression values were mean-centered (genewise), and cosine similarity was used as the distance measure.

C  Genes induced upon TGFβ treatment of TA organoids, which are differentially expressed between TA and SSA samples, display significantly higher expression levels in CMS4-SSAs compared with CMS1-SSAs (*n* = 8 for CMS1-SSAs and *n* = 4 for CMS4-SSAs. The horizontal lines represent the mean, and the error bars display the SD. *P*-values are based on unpaired two-tailed Student's *t*-tests.).

D  Gene expression-based classification reveals that SSAs could progress to CMS1 and CMS4 tumors. The data presented in this manuscript suggest that high activity of the TGFβ signaling pathway directs these precursor lesions to the CMS4 of CRC.

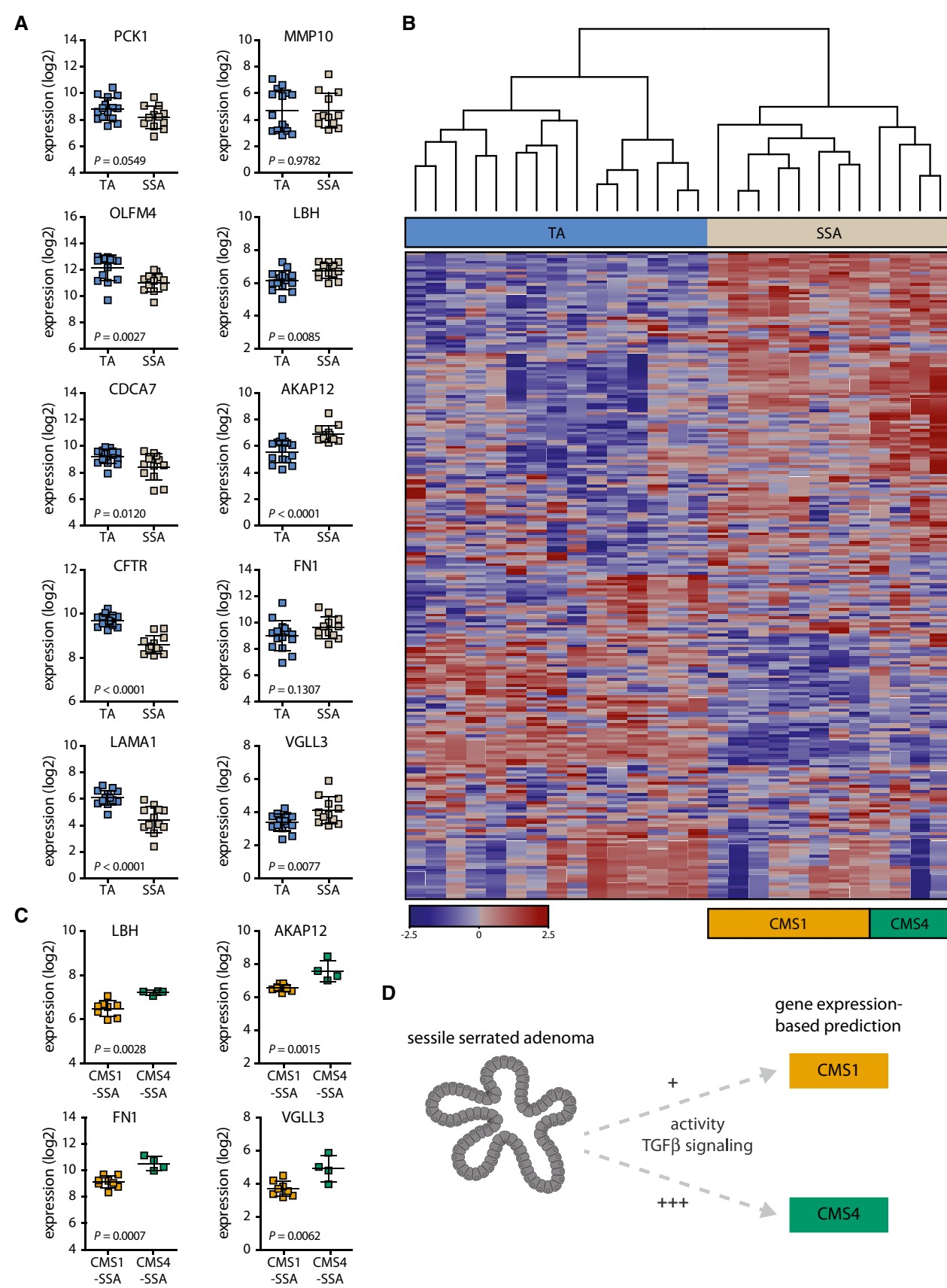

**Figure 4.**

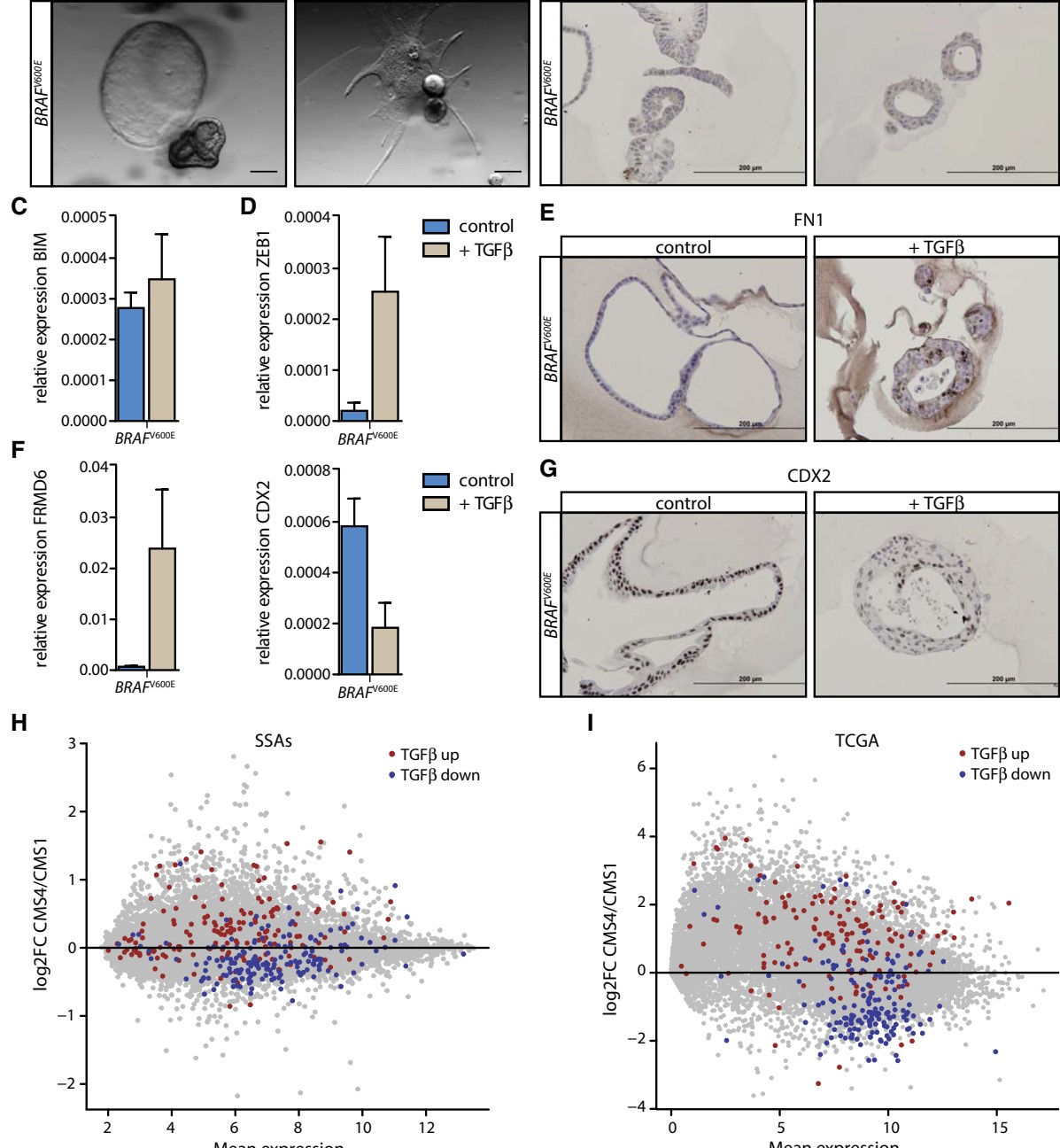

**Figure 5.  A *BRAF*^V600E-mutant organoid culture serves as model for the serrated path to CRC.**

A     An organoid culture genetically engineered using the CRISPR-Cas9 system to carry a *BRAF*^V600E mutation represents a model for the serrated path to CRC. Upon TGFβ treatment, these culture gains a mesenchymal appearance (scale bars: 200 μm).

B, C   TGFβ stimulation does not induce (B) cleaved Caspase-3 or (C) BIM expression (*n* = 6) in the *BRAF*^V600E-mutated organoid culture (scale bars: 200 μm; error bars represent SD).

D, E   Markers of the EMT program, (D) ZEB1 (*n* = 6) and (E) FN1 are strongly induced in the TGFβ-treated condition (scale bars: 200 μm; error bars represent SD).

F, G   FRMD6 is upregulated and CDX2 expression is reduced both on RNA (*n* = 6) as well as on protein level following TGFβ stimulation (scale bars: 200 μm; error bars represent SD).

H     TGFβ target genes downregulated upon TGFβ treatment of the *BRAF*^V600E-mutant organoid culture (TGFβ down) are lower expressed in CMS4-SSAs compared to CMS1-SSAs and genes induced by TGFβ treatment (TGFβ up) are more strongly expressed in CMS4-SSAs than in CMS1-SSAs (*n* = 4 for CMS4-SSAs and *n* = 8 for CMS1-SSAs; FC = fold change).

I     The differential expression of TGFβ target genes observed for CMS1-SSAs and CMS4-SSAs is maintained at the carcinoma stage for CMS4 versus CMS1 *BRAF*-mutant CRCs of the TCGA dataset (*n* = 4 for *BRAF*-mutant CMS4 and *n* = 34 for *BRAF*-mutant CMS1 CRC samples; FC = fold change).

and offer a possible explanation as to how the same pre-neoplastic lesion might spawn CRCs belonging to the most favorable and the most dismal prognosis subtype.

## Discussion

The roles of TGFβ in CRC tumorigenesis are manifold and not undisputed in literature. While some studies indicate that the main target of this growth factor is the tumor stroma due to unresponsiveness of the malignant epithelium (Calon et al, 2012), others report that active TGFβ signaling can indeed be detected in epithelial tumor cells (Brunen et al, 2013). Notwithstanding the exact mechanism of action, increased levels of TGFβ pathway activity are predictive of prognosis and point to the presence of metastatic lesions (Friedman et al, 1995; Robson et al, 1996; Tsushima et al, 2001; Calon et al, 2012; Brunen et al, 2013). This can in part be explained by the ability of TGFβ to induce EMT, a process that is linked to poor disease outcome (Shioiri et al, 2006; Spaderna et al, 2006). Two recent reports enforce this notion and indeed associate the EMT program with dismal outcome. The use of mouse models allowed the authors of these studies for the first time to follow the EMT process in vivo and suggests that the mesenchymal phenotype is not related to metastatic dissemination, but to chemoresistance (Fischer et al, 2015; Zheng et al, 2015). While the mechanism by which EMT confers poor prognosis is thus a matter of debate, it has been firmly established that a mesenchymal cancer phenotype is linked to dismal disease outcome. Indeed, gene expression-based classification of CRC samples identified a mesenchymal subtype associated with poor clinical outcome (Guinney et al, 2015). Despite the fact that the CMSs of CRC cannot be identified based on unifying molecular markers currently used in the clinic, they represent biologically homogeneous groups, allowing the speculation that common underlying drivers are responsible for installing specific biological programs. The activation of the EMT program is a hallmark feature of the mesenchymal CMS4 of CRC and the TGFβ signaling pathway—a known inducer of this process (Moustakas & Heldin, 2007)—is predicted to be active in CMS4 tumors (Guinney et al, 2015). Therefore, we have herein elucidated the effect of TGFβ on subtype affiliation at the early stage of tumor development. We show that the genetic background of pre-neoplastic lesions dictates the dominating response to this signaling molecule, changing it from a largely apoptotic response in WNT pathway-activated TAs to a dominant EMT response in mitogen-activated protein kinase (MAPK) pathway mutant cells. In addition, our data show that TGFβ influences subtype affiliation. Our data demonstrate that— already in premalignant lesions—TGFβ can induce the EMT program. The severity of the simultaneously occurring apoptotic response is strongly dependent on the genetic background: Whereas it dominates in TAs, wild-type organoids and cultures with mutations in the MAPK pathway are protected from apoptosis. Mutations in the MAPK pathway in the form of activating KRAS mutations occur during tumor progression in the classical path of CRC development (Fearon & Vogelstein, 1990). In contrast, activation of this pathway by the BRAF$^{V600E}$ mutation is an initiating mutation in the serrated neoplasia pathway (Leggett & Whitehall, 2010; Snover, 2011). Since early SSA lesions can therefore exploit TGFβ signaling without dying by apoptosis, this growth factor is able to direct

subtype affiliation early in the developmental path of SSAs. Indeed, our data indicate that high levels of TGFβ signaling can poise SSA lesions to develop to the mesenchymal CMS4 of CRC. Importantly, TGFβ stimulation led to downregulation of CDX2, whose expression is low or absent in poor-prognosis, mesenchymal CRCs (De Sousa E Melo et al, 2013). This observation has been re-enforced by a recent study (Dalerba et al, 2016); however, it is not clear how and at what stage in tumor progression CDX2 expression is lost. Our data suggest that TGFβ signaling downregulates CDX2 expression, possibly early in tumor progression. Furthermore, the induction of an EMT phenotype, which has been linked to the stem-like and thus undifferentiated state (Mani et al, 2008), can be stimulated by TGFβ (Taube et al, 2010) and could therefore explain both low levels of CDX2 expression and the immature phenotype of these cancer cells.

In contrast to apoptosis, the induction of growth arrest is not influenced by the mutation status of the organoid cultures. However, it is important to note that the genetically engineered BRAF$^{V600E}$-mutant organoid culture did not display a hypermethylation phenotype that is thought to be an early event in the serrated path to CRC (Park et al, 2003; Kambara et al, 2004). Indeed, as reported before, we observed methylation of the CDKN2A locus (encoding for p16$^{INK4A}$/p19$^{ARF}$) in a panel of BRAF$^{V600E}$-mutant SSAs (data not shown), which is most likely installed to overcome oncogene-induced senescence (Carragher et al, 2010). We speculate that the epigenetic silencing of p16$^{INK4A}$/p19$^{ARF}$ expression might allow the cells to also circumvent the TGFβ-induced growth arrest and progress to a malignant stage. This hypothesis warrants further testing with appropriate genetically engineered cultures.

It will be of interest to further investigate whether the premalignant lesions progress to CRC while maintaining their TGFβ-responsiveness or whether TGFβ stimulation early in tumor development installs a more aggressive phenotype without the constant need of restimulation during progression. BRAF$^{V600E}$-mutant CMS4 tumors of the TCGA dataset display higher levels of TGFβ pathway activation based on gene expression compared with CMS1 BRAF$^{V600E}$-mutant CRCs, suggesting that the distinction installed early in tumor development is maintained at the carcinoma stage. This long-term commitment may be explained by the fact that TGFβ can induce DNA hypermethylation (Papageorgis et al, 2010; Davalos et al, 2012). DNA methylation represents a metastable mechanism that can be inherited by daughter cells, allowing for stabilization of specific phenotypes—such as EMT—without the need for continuous signaling input. However, the organoid cultures used in this study did not induce methylation of the miR-200 promoter regions upon TGFβ treatment, which might be a reflection of in vitro culture conditions. Specific requirements for the induction of DNA hypermethylation in vivo might not be mimicked by in vitro culture systems lacking microenvironmental components such as cancer-associated fibroblasts (CAFs) and distinct extracellular matrix components. Alternatively, even though CRCs frequently contain mutations in TGFβ pathway components (Markowitz et al, 1995; Fleming et al, 2013), cancer cells can maintain responsiveness to the TGFβ signaling molecule. It has, for instance, recently been shown that instead of an overall insensitivity to TGFβ, SMAD4 inactivation leads to a shutdown of selective programs that would be opposing malignant progression, such as apoptosis (David et al, 2016). Continuous TGFβ signaling during tumor progression could therefore install subtype affiliation early in the adenoma–carcinoma

sequence and subsequently remain necessary to maintain it throughout development.

To study the full response of SSAs to TGFβ, their dependence on this signaling molecule, and the dominating phenotype upon stimulation, organoid cultures derived from these lesions are indispensable. Thus far, we and others have not been able to establish organoids from serrated adenomas (IJspeert *et al*, 2015). However, the recent advances in genome editing using the CRISPR-Cas9 system have made it possible to engineer genetically defined organoid cultures (Drost *et al*, 2015). Herein, we used this technique to generate a *BRAF*^V600E^-mutant organoid culture to serve as an *in vitro* model system for early serrated lesions. The fact that these organoids have not progressed beyond the initial hit in the *BRAF* gene, yet, allowed the elucidation of early events in the serrated neoplasia pathway. Combining data from this model system with gene expression-based information from SSA and TA samples prompts us to hypothesize that TGFβ plays a role in the serrated pathway to CRC.

Our data indicate that high activity of the TGFβ signaling pathway can direct SSAs to the CMS4 of CRC, yet, the source of this signaling molecule remains to be determined. Of note is the fact that poor-prognosis CRCs display a stroma-rich environment (Calon *et al*, 2015; Isella *et al*, 2015). One of the main stromal components are CAFs, which have been described to secrete the TGFβ signaling molecule (Calon *et al*, 2014). Additionally, the abundant presence of CAFs might provide a favorable environment for the premalignant cells to overcome the TGFβ-induced growth arrest and to progress to a malignant, and eventually to a metastatic state. Whereas TGFβ is detrimental to the epithelium of TA precursor lesions, SSAs might benefit from TGFβ pathway activation, priming them to adapt a more aggressive phenotype and directing these precursor lesions toward the mesenchymal, poor-prognosis CRC subtype.

# Materials and Methods

### Sample collection

Human tissues were obtained in accordance with the legislation in the Netherlands. Collection of normal and adenomatous material from the colon was approved by the Medical Ethical Committee (normal tissue: 2014_178; adenomas: MEC 09/146 and MEC 05/071; Academic Medical Center (AMC), Amsterdam). Tissue was collected following written informed consent of patients, and the experiments conformed to the principles set out in the WMA Declaration of Helsinki and the Department of Health and Human Services Belmont Report.

### Processing of TA and SSA samples

For RNA and gDNA extraction from patient material, $5 \times 20$ μm frozen tissue sections were cut using a cryotome and stored at −80°C until further use. Fifteen TAs were obtained from 10 familial adenomatous polyposis (FAP) patients, and 12 sessile serrated adenomas (SSAs) were collected from 4 serrated polyposis syndrome (SPS) patients. We confirmed the nature of each polyp by H&E staining. RNA was extracted using the miRNAeasy micro kit

(Qiagen). Microarrays of these samples were performed as described below, and the resulting gene expression profiles were analyzed in combination with those described in (GSE45270, De Sousa E Melo *et al*, 2013).

### Organoid isolation, culture, and treatment

Human adenoma cultures TA1 and TA2 were derived from two independent polyps of one patient and generated and maintained as previously described (Sato *et al*, 2011; Prasetyanti *et al*, 2013). TA3-TA5 organoid cultures were derived from polyps obtained from three additional patients and established by cutting the polyps into small pieces using tweezers and a scalpel. After washing, the pieces were plated in growth factor-reduced BD matrigel matrix (BD Biosciences, further referred to as matrigel) and mechanically dissociated upon growth to propagate viable organoids [the naming of the organoid cultures (TA1–TA5) is independent of the TA adenoma samples used for gene expression array (TA2–TA16)]. Please see Appendix Supplementary Methods for propagation and medium composition.

Normal colon tissue was obtained from resection material of CRC patients from a part of the mucosa $\geq 10$ cm apart from the cancerous tissue. Normal colon organoid cultures were isolated as previously described (Sato *et al*, 2011), and for a detailed description of isolation, propagation, and medium composition, please see Appendix Supplementary Methods.

The organoid cultures were treated with 5 ng/ml recombinant human TGF-β 1 (PeproTech 100-21) for 5 days, and medium was refreshed after 3 days. Control refers to TA culture medium + 1 μM A83-01 + TGFβ dissolvent, and to normal colon culture medium + TGFβ dissolvent (regular culture medium already contains A83-01).

Organoid cultures were mycoplasma negative, as determined by testing for mycoplasma contamination every 4–6 weeks.

Phase contrast pictures were taken with a Zeiss Axiovert 200M fluorescence microscope.

### Generation of the human *BRAF*^V600E^-mutant colon organoid culture

The *BRAF*^V600E^ mutation was introduced by homologous recombination in human colon organoids using a targeting vector containing a puromycin-resistance cassette flanked by 5′- (including the *BRAF*^V600E^ sequence) and 3′-homology arms. The puromycin-resistance cassette was targeted to the intron downstream of *BRAF* exon 15. Organoids were cotransfected with a sgRNA targeting *BRAF* (target sequence 5′-TAG CTA CAG TGA AAT CTC GAT GG), Cas9 endonuclease, and the targeting plasmid as described (Drost *et al*, 2015). Growth medium was exchanged with medium containing 1 μg/ml puromycin 3 days after transfection. For clonal expansion, single puromycin-resistant organoids were picked. Genotyping was performed as previously described (Drost *et al*, 2015). Primer sequences were as follows: *BRAF*-genotyping-forward: 5′-GTT AGT CAT GGG AAA GCT TC; *BRAF*-genotyping-reverse: 5′-GCC TCC CCT ACC CGG TAG AAT T. *BRAF*^V600E^-mutant organoids were cultured in normal colon culture medium as described in the Appendix Supplementary Methods and dissociated every 7 days, and the medium was refreshed every 3–4 days.

## RNA extraction and quantitative real-time PCR

For RNA extraction from organoid cultures, matrigel was destroyed mechanically and organoids were resuspended in cell recovery solution (BD Biosciences) and incubated on ice. Cells were pelleted, and RNA was extracted using the NucleoSpin RNA II kit from Macherey-Nagel.

To determine gene expression levels by qRT–PCR, total RNA was reverse-transcribed to cDNA using Superscript III following the manufacturer's protocol (Invitrogen). qRT–PCR was performed using SYBR Green (Roche) and a Roche Light Cycler 480 II in accordance with the manufacturer's instructions. All obtained values were normalized to the expression of β-actin, and normalization to B2M yielded similar results. Please see the Appendix Supplementary Methods for primer sequences.

## Western blotting

To prepare protein lysates, matrigel was mechanically disrupted and resuspended in PBS. Organoids were pelleted and washed once with PBS. Organoids were resuspended in 1X reducing sample buffer, and lysates were sonicated and boiled for 5 min at 85°C. Protein concentration was determined using the Protein Quantification Assay from Macherey-Nagel. Ten micrograms of protein was loaded on 4–15% Mini-PROTEAN® TGX™ Precast Protein Gels (Bio-Rad) and transferred to a hydrophobic, microporous, polyvinylidene difluoride (PVDF) membrane with a pore size of 0.2 μm (Roche). Membranes were blocked for 1 h in 5% bovine serum albumin in PBS + 0.2% Tween-20 (Sigma) and subsequently incubated in primary antibody dilution in 5% bovine serum albumin in PBS + 0.2% Tween-20 at 4°C overnight [anti-phospho-Smad2 (Ser465/467) (138D4) (1:2,000, Cell Signaling 3108); anti-Smad2 (L16D3) (1:2,000, Cell Signaling 3103)]. The secondary antibody in 5% bovine serum albumin in PBS + 0.2% Tween-20 [anti-rabbit-HRP (1:2,000 Cell Signaling 7074); anti-mouse-HRP (1:10,000, Southern Biotech 1070-05)] was applied for 1 h at room temperature. After washing, the membrane was developed using Lumi-Light$^{PLUS}$ Western Blotting Substrate (Roche), and signal was detected with the ImageQuant LAS4000 (GE Healthcare Life Sciences). Washing steps were performed using PBS + 0.2% Tween-20.

## Immunohistochemistry

For immunohistochemical stainings, the matrigel cultures were washed twice with PBS and fixed using 4% paraformaldehyde (Klinipath) overnight at 4°C. Subsequently, paraformaldehyde was replaced with 70% ethanol. After an incubation period of 30 min at RT, cells were incubated for 30 min at RT in 96% ethanol containing hematoxylin to visualize the organoids. Dehydration was continued with 100% ethanol followed by xylene each twice for 30 min at RT. Subsequently cells were incubated in paraffin at 60°C twice for 30 min, followed by embedding in paraffin.

Four-micrometer sections of formalin-fixed paraffin-embedded organoids were used for immunohistochemistry.

CDX2, SMAD4, FN1: After rehydration, antigen retrieval was performed using a 10 mM sodium citrate buffer at pH 6.0 (Vector Laboratories) for 20 min at 98°C. Endogenous peroxidase was blocked with 3% H$_2$O$_2$ (VWR) in PBS for 15 min at RT followed by incubation in ultraV block (Immunologic) for 10 min at RT. Sections were incubated in primary antibody dilution at 4°C overnight [anti-CDX2 (1:100, BioGenex, clone CDX2-88, MU392A); anti-SMAD4 (B-8) (1:400, Santa Cruz Biotechnology sc-7966); anti-Fibronectin Clone 10/Fibronectin (1:100, BD Biosciences 610077)]. Subsequently, post-antibody blocking (Immunologic) was added for 20 min at RT followed by poly-HRP-anti-mouse/rabbit/rat IgG (Immunologic) for 30 min at RT. Washing steps were performed using PBS. Sections were incubated in Bright DAB solution (Immunologic), rinsed in dH$_2$O, and counterstained using hematoxylin (Klinipath). After dehydration, slides were mounted using Pertex (HistoLab). Images were taken with a Leica TCS-SP2.

ZEB1: Same as the immunohistochemistry protocol for CDX2, SMAD4, FN1 above, with the exception of antigen retrieval: performed using an EDTA-based buffer at pH 9.0 (Vector Laboratories) for 20 min at 98°C. Sections were incubated in primary antibody dilution at 4°C overnight [anti-ZEB1 (1:400, Sigma HPA027524)].

Cleaved Caspase-3 (Asp175): After rehydration, antigen retrieval was performed using a 10 mM sodium citrate buffer at pH 6.0 (Vector Laboratories) for 30 min at 95°C. Endogenous peroxidase was blocked with 3% H$_2$O$_2$ (VWR) in methanol for 20 min at RT followed by incubation in ultraV block (Immunologic) for 5 min at RT. Sections were incubated in primary antibody dilution at 4°C overnight [anti-cleaved Caspase-3 (Asp175) (1:200, Cell Signaling 9661)]. Subsequently, poly-HRP-anti-mouse/rabbit/rat IgG (Immunologic) was added for 60 min at RT. Washing steps were performed using PBS + 0.05% Tween-20. Sections were incubated in Bright DAB solution (Immunologic), rinsed in dH$_2$O, and counterstained using hematoxylin (Klinipath). After dehydration, slides were mounted using Pertex (HistoLab). Images were taken with a Leica TCS-SP2.

## Mutation analysis

gDNA was extracted from the TA organoid cultures and from matching samples of 18 patient-derived adenoma samples for which gene expression profiles were derived using the NucleoSpin Tissue kit from Macherey-Nagel. BRAF, KRAS, NRAS, and PIK3CA mutations were analyzed using two multiplex PCRs as described before (Lurkin et al, 2010). The TA1 organoid culture was subjected to PCR followed by direct Sanger sequencing to confirm the KRAS mutation. Amplification of exon 2 of the KRAS gene was carried out in a 25 μl total reaction volume consisting of 20 ng gDNA, 12.5 μl Reddymix (Thermo Scientific), 1 μl forward and reverse primers (10 μM), and 6.5 μl H$_2$O. Samples were subjected to 5 min 95°C, 40 cycles of 30 s 95°C, 30 s 55°C, 1 min 30 s 72°C, followed by 5 min 72°C. One microliter of PCRs was subsequently sequenced using BigDye Terminator 3.1 (BDT, Applied Biosystems). Primer sequences: KRAS_exon2-forward: 5′-GTG TGA CAT GTT CTA ATA TAG TCA; KRAS_exon2-reverse: 5′-GAA TGG TCC TGC ACC AGT AA. Please see the Appendix Supplementary Methods for analysis of the BRAF mutation and CIMP status of patient-derived adenoma samples.

## Nicoletti assay

Matrigel was destroyed mechanically, and organoids were resuspended in cell recovery solution (BD Biosciences) and incubated on ice. Cells were pelleted and resuspended in Nicoletti buffer (0.1% sodium citrate (w/v) and 0.1% Triton X-100 (v/v) in deionized

water pH 7.4, supplemented with 50 µg/ml propidium iodide before use) (Nicoletti *et al*, 1991). After incubation at 4°C, PI staining of nuclei was analyzed using flow cytometry (FACS Canto). Measurements were performed in triplicate for each cell line and condition, and the experiment was repeated at least three independent times.

**Microarray**

Microarrays of TA, normal, and *BRAF*^V600E^-mutant organoid cultures as well as patient-derived TA and SSA samples were performed using the GeneTitan™ MC system from Affymetrix according to the standard protocols of the Cologne Center for Genomics (CCG), University of Cologne, Germany. Three TA, two normal colon, and the genetically engineered *BRAF*^V600E^-mutant organoid cultures were included in the microarray, and the replicates included in the microarray were as follows: TA1 control *n* = 1, + TGFβ *n* = 1; TA2 control *n* = 2, + TGFβ *n* = 4; TA3 control *n* = 2, + TGFβ *n* = 1; N2 and N3 control *n* = 1, + TGFβ *n* = 1; *BRAF*^V600E^ control *n* = 3, + TGFβ *n* = 2; patient-derived TA *n* = 9 and SSA *n* = 7. Only samples with an RNA integrity number (RIN) above 7 were included. Microarray data can be viewed online under GEO accession number GSE79462 [comprising microarrays of patient-derived adenomas (GSE79460) and organoid cultures (GSE79461)].

**Gene expression data**

The microarray data were normalized and summarized using robust multiarray analysis (rma) (Irizarry *et al*, 2003), and batch effects were removed using the Combat algorithm (Johnson *et al*, 2007) as implemented in the sva R package (R package version 3.18.0). After normalization, the probe sets were annotated using the hgu133-plus2.db annotation package [Affymetrix Human Genome U133 Plus 2.0 Array annotation data (chip hgu133plus2) R package version 3.1.3]. In case of multiple probe sets interrogating a specific gene, the probe set with the highest mean intensity was selected as representative for that gene.

**TGFβ signature**

A TGFβ signature was established by comparing gene expression following TGFβ and control treatment in three TA organoid cultures. Genes differentially expressed between TGFβ and control treatment were identified using the limma R package (Ritchie *et al*, 2015) in which both the treatment and organoid culture-specific effects were modeled. Genes with a *P*-value < 0.01 (adjusted for multiple testing using the Benjamini & Hochberg method) and an absolute log2 fold change > 3 were selected to be part of the TGFβ signature (231 genes). The same approach was used to determine a TGFβ signature in the genetically engineered *BRAF*^V600E^-mutant organoid culture (334 genes).

**Gene set enrichment analysis**

Gene set enrichment analyses (GSEAs) (Mootha *et al*, 2003; Subramanian *et al*, 2005) were performed using the default settings of the Broad Institute's web application tool (http://www.broadinstitute.org/gsea/index.jsp). *P*-values indicating the significance of enrichment were estimated by 1,000 permutations. The following

**The paper explained**

**Problem**
Colorectal cancer (CRC) is a heterogeneous disease, which hampers accurate prognostication and the efficacy of adjuvant therapy. Recently, gene expression-based classification has allowed the identification of four consensus molecular subtypes (CMSs) of CRC and samples belonging to each group are defined by common biological programs. However, to date it is not clear what is responsible for driving the development to one specific CMS. Of special interest is the mesenchymal CMS4 as patients in this group present with dismal prognosis. Gene expression data point to a role of transforming growth factor-β (TGFβ) in this subset of CRC and in its associated premalignant lesion, the SSA.

**Results**
We studied the effect of TGFβ at the premalignant stage of CRC development. We made use of organoid cultures from normal colon epithelium, (TAs; precursor lesions of the classical, good-prognosis CMS), and of an organoid culture genetically engineered to carry a *BRAF*^V600E^ mutation, which frequently occurs early in the serrated path to CRC. Our data show that an apoptotic phenotype prevailed in TA organoids upon TGFβ treatment, whereas cultures from normal epithelium and from material carrying an activating mutation in the MAPK pathway were protected from apoptosis induction. TGFβ stimulation of organoid cultures led to the induction of a mesenchymal phenotype and they adapted their gene expression profiles to those of SSAs and the mesenchymal CMS4.

**Impact**
Our data on premalignant lesions of CRC indicate that distinct genetic backgrounds respond differentially to TGFβ treatment and that SSAs might benefit from TGFβ stimulation. We hypothesize that high levels of TGFβ signaling activity install a mesenchymal phenotype in these distinct CRC precursor lesions, directing them to the mesenchymal, poor-prognosis CRC subtype.

publicly available signatures were used: an apoptosis signature (KEGG) and two EMT signatures (Taube *et al*, 2010; Gröger *et al*, 2012). Furthermore, we applied a signature comprising genes upregulated in CMS4 compared to CMS2 + 3 tumor samples of the AMC-AJCCII-90 dataset (log2FC > 2, GSE33113), the 500 most highly expressed genes in SSA compared with TA samples (GSE45270), a signature of genes induced upon TGFβ treatment in TA organoids (log2FC > 3; described above), and a signature of genes induced upon TGFβ treatment in the *BRAF*^V600E^ organoid culture (log2FC > 3; described above).

**Molecular subtype classification of adenomas**

Microarray data for the two batches of adenoma samples, newly generated and from (De Sousa E Melo *et al*, 2013), were first separately normalized and summarized using the RMA method (Irizarry *et al*, 2003). Non-biological effects between the two batches were detected using principal component analysis and were corrected using ComBat (Johnson *et al*, 2007). The corrected expression profiles were collapsed from probe sets to unique genes and median-centered across all samples. The classifier (Guinney *et al*, 2015) developed by the colorectal cancer subtyping consortium (CRCSC) was then employed to classify adenoma samples into consensus molecular subtypes. Only samples that could be classified with a posterior probability of ≥ 0.5 were included in the subsequent

analyses and their gene expression profiles are available at GEO: GSE45270 (TA: GSM1100484-GSM1100489 and SSA: GSM1100491-GSM1100495) and GSE79460.

### Survival analysis

The association of the TGFβ signature with recurrence-free survival was tested using two publicly available colon tumor gene expression datasets [GSE33113 (De Sousa E Melo *et al*, 2011) and TCGA-COAD Illumina HiSeq and genome analyzer (GA) (http://cancergenome.nih.gov/)]. Stage IV patients were excluded from the TCGA dataset for this analysis. The samples in both datasets were divided into TGFβ signature low and high groups using the Pearson correlation with the 231-gene TGFβ signature derived from the TA organoid cultures. Samples with a correlation > 0.1 were assigned to the TGFβ high group and samples with a correlation < −0.1 to the TGFβ low group. The survival distributions of the two groups were compared with the log-rank test.

### Statistics

The log-rank test was used to compare the RFS of different groups shown in the KM plots. The Pearson's correlation test makes use of a *t* distribution with number of observations -2 degrees of freedom to determine a *P*-value. For all other comparisons, unpaired two-tailed Student's *t*-tests were used. If not otherwise indicated, mean ± SD is depicted. Experiments with TA and *BRAF*[V600E]-mutant organoid cultures were performed at least three independent times, of which one representative experiment is depicted in the figures (if not otherwise indicated in the figure legend). Due to the limited life span of normal colon organoid cultures, experiments with N1-N3 organoids could not all be performed three independent times (N3 qRT–PCR *n* = 3, N3 IHC *n* = 2, and N1 + N2 qRT–PCR *n* = 1). A *P*-value of below 0.05 was considered significant.

**Expanded View** for this article is available online.

### Acknowledgements
This work was supported by an AMC Graduate School PhD Scholarship (E.F.), NWO-ZonMw VENI grant (J.D., 91614138), and Dutch Cancer Society grants UvA2013-6331, UvA2014-7245, and UvA2015-7587 (J.P.M.).

### Author contributions
EF, JFL, FDSEM, and PRP collected patient material, isolated normal and TA organoid cultures, and performed experiments; JD generated the *BRAF*[V600E]-mutant organoid culture; SRvH and XW performed bioinformatical analyses; MJ, JEGI, CJMvN, and ED provided tissue and pathological assistance; MF and PN conducted microarrays; EF, LV, and JPM designed the experiments and analyzed the data; EF and JPM wrote the manuscript; and LV, HC, and JPM supervised the project.

### Conflict of interest
The authors declare that they have no conflict of interest.

### For more information
http://www.ncbi.nlm.nih.gov/geo/query/acc.cgi?acc=GSE33113
http://www.ncbi.nlm.nih.gov/geo/query/acc.cgi?acc=GSE45270
http://cancergenome.nih.gov/cancersselected/colorectaladenocarcinoma

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
