## [Review Process File · EMBO Molecular Medicine]

TGF β signaling directs serrated adenomas to the mesenchymal colorectal cancer subtype

Evelyn Fessler, Jarno Drost, Sander R. van Hooff, Janneke F. Linnekamp, Xin Wang, Marnix Jansen, Felipe De Sousa E Melo, Pramudita R. Prasetyanti, Joep E. G. IJspeert, Marek Franitza, Peter Nürnberg, Carel J. M. van Noesel, Evelien Dekker, Louis Vermeulen, Hans Clevers, and Jan Paul Medema

Corresponding author: Jan Medema, Center for Experimental Molecular Medicine

Review timeline:

Submission date:	05 January 2016
Editorial Decision:	05 February 2016
Revision received:	24 March 2016
Editorial Decision:	13 April 2016
Revision received:	19 April 2016
Accepted:	20 April 2016

Transaction Report:

Editor: Roberto Buccione

1st Editorial Decision

05 February 2016

Thank you for the submission of your manuscript to EMBO Molecular Medicine. We have now heard back from the three Reviewers whom we asked to evaluate your manuscript.

Although Reviewers 1 and 3 are generally positive, Reviewer 2 is much more reserved and raises issues that are of a fundamental nature. I will not dwell into much detail, but I would like to highlight the main points.

The main concern expressed by Reviewer 2 is connected to the fact that to overcome the lack of a bona fide SSA organoid model, your strategy is to express BRAFV600E in normal mucosa organoids. However, due to a number of reasons directly connected to this choice, which are well explained and that you are surely aware of, the interpretation of the data is fraught with caveats weakening the conclusions. I also notice that, notwithstanding a more positive stance, Reviewer 1 and to some extent also Reviewer 3 do, as a matter of fact, alert to the same basic issues. Reviewer 2 also raises a number of other concerns, which are mostly interconnected to the same basic perceived flaw. I should add that I also agree that the abstract and the discussion are unduly unclear. In essence, although admittedly his/her style is a bit cursory, Reviewer 2's comments tend to supersede the positive ones and actually confirm my own concerns when deciding whether to send the manuscript out for peer review

As you know, we also ask the reviewers to cross-comment on each other's evaluations. Reviewers 1

and 3 came to agree to a good extent that the points raised are indeed valid. Clearly, although many of the claims could be simply very much toned down, this would devalue the actual conclusions due to lack of sufficient support. Therefore it becomes absolutely necessary to also experimentally address the connection between SSA and CMS4 CRCs. To better "explain" such differences will clearly not be sufficient.

In conclusion, after further internal discussion, we reached the decision to allow you the opportunity to address the criticisms in detail in a major revision, with the understanding that the Reviewers' concerns must be addressed with additional experimental data where appropriate and that acceptance of the manuscript will entail a second round of review with no guarantee of final acceptance given the issues at hand.

I understand that if you do not have the required data available at least in part, to address the above, this might entail a significant amount of time, additional work and experimentation and might be technically challenging, I would therefore understand if you chose to rather seek publication elsewhere at this stage. Should you do so, we would welcome a message to this effect.

Please note that it is EMBO Molecular Medicine policy to allow a single round of revision only and that, therefore, acceptance or rejection of the manuscript will depend on the completeness of your responses included in the next, final version of the manuscript.

EMBO Molecular Medicine now requires a complete author checklist (<http://embomolmed.embopress.org/authorguide#editorial3>) to be submitted with all revised manuscripts. Provision of the author checklist is mandatory at revision stage; The checklist is designed to enhance and standardize reporting of key information in research papers and to support reanalysis and repetition of experiments by the community. The list covers key information for figure panels and captions and focuses on statistics, the reporting of reagents, animal models and human subject-derived data, as well as guidance to optimise data accessibility. This checklist especially relevant in this case given the issues raised with respect to statistical treatment and animal numbers.

As you know, EMBO Molecular Medicine has a "scooping protection" policy, whereby similar findings that are published by others during review or revision are not a criterion for rejection. However, I do ask you to get in touch with us after three months if you have not completed your revision, to update us on the status. Please also contact us as soon as possible if similar work is published elsewhere.

***** Reviewer's comments *****

Referee #1 (Comments on Novelty/Model System):

Model adequate though association to CMS4 stretched as mentioned in remarks

Referee #1 (Remarks):

Fessier et al. describe a serrated-like phenotype induced by TGFB in normal and adenomatous colon organoids. This is a worthy work shedding light on diverse effects of TGFB signalling, as well as genesis of serrated colon tumors.

The paper is logical and clear and a pleasure to read, and the story fits nicely the previous knowledge and adds to it. The one concern I have is that interesting effects of TGFB are in particular observed in the BRAF mutant organoid model, and the findings are then extrapolated to CMS4 type cancers (as defined by Guinney et al 2015) and their characters such as EMT & TGFB activation; but it is not really discussed that most BRAF mutations occur in CRCs other than type CMS4 (see eg Guinney et al. Figure 3). Thus, how well the two actually come together in real life; the observations in the BRAF mutant organoid model, and CMS4 tumor characters?

Detailed:

Page 7 and 8 please provide information of the mutation status at organoid TA1 earlier, the reader is a bit lost why this behaved differently and in the current version needs to wait for the explanation.

Page 10 "genes present in a poor prognosis signature associated with TGF β signalling were highly enriched in samples belonging to the mesenchymal CMS and in SSA lesions" this sounds like circular reasoning.

Page 16 CRC tumors sounds odd; instead CRCs or colorectal tumors plz.

Referee #2 (Comments on Novelty/Model System):

An ideal, serrated adenoma organoid model would be that established directly from SSAs.

Referee #2 (Remarks):

The title of this manuscript claims that TGF β induces a serrated-like phenotype in colon organoids.

The serrated phenotype, i.e., a jagged contour of intestinal glands, is not visible in the organoids shown in the figures of this manuscript. In fact, the authors have twice written that they have not been able to obtain organoid from cells of serrated colon mucosa (i.e., from SSAs). (Btw, this statement is accompanied by a reference to a review article in which experiments showing that these attempts were unsuccessful are not shown).

To overcome this problem, the authors have expressed the mutant BRAFV600E in organoids derived from normal mucosa, since BRAF is frequently mutated in SSAs. This is not a suitable model since SSAs likely do not carry only a BRAFV600E mutation, but several other alterations that are likely absent in such organoids.

The "phenotype" the authors meant in the title was evidently related to the gene expression signature triggered by exposing organoids to TGF β : there was some overlap between the signature they previously found in 6 SSAs (Melo 2013) and that that they have found now by treating all types of organoids (normal mucosa, TA and BRAFV600E organoids) with TGF β . The only difference was that the KRAS mutated TA1 and the BRAFV600E organoids stopped growing, while TA organoids went to apoptosis. Therefore, it seems that the presence of such mutations -whether in KRAS or BRAF- changes the cellular response to TGF β , from an apoptotic to a growth arrest phenotype. However, KRAS mutations are rare in SSAs, therefore the authors cannot conclude that these phenotypes are typical of the serrated pathway.

In addition, finding some overlap among transcriptomic signatures does not justify making a direct pathogenetic link between different types of tumors, like SSA and CMS4 colon cancers. Although I cannot access the accuracy of the enrichment analysis performed in this study, it is a dangerous game to make conclusions by comparing gene expression data from several different studies, even one from ovarian cancers. Even more dangerous is then to correlate these findings with what is generally known on the prognosis of such tumors. This problem emerges clearly from sentences like these:

page 11 and 12: "These data indicate that precursor lesions from distinct backgrounds respond differentially to TGF stimulation early in tumor development. The apoptotic response dominated in classical TAs, yet, in both TA and BRAFV600E-mutated organoids a robust EMT response was observed upon TGF treatment. In the case of TA organoids, this seems to be a selection-based response, whereas the majority of BRAFV600E-mutated organoids is able to induce the EMT process. Thus, BRAFV600E-mutant SSAs might benefit from TGF pathway activation directing these precursor lesions to a more aggressive phenotype, marked by the induction of the EMT program, toward the mesenchymal, poor prognosis CRC subtype."

First sentence: The authors have not investigated precursor lesions from different background; they have studied organoids in vitro.

Second and third sentences: they first write that both TA and BRAFV600E organoids had a robust EMT response (second sentence), but conclude the third sentence with "whereas the majority of BRAFV600E-mutated organoids is able to induce the EMT process". Also, what is a "selection-

based response"?

Forth sentence: why would the SSAs benefit from the TGFbeta pathway activation? because they would stop growing instead of dying via apoptosis? Also, the direct relationship between SSAs and the poor prognosis, mesenchymal colon cancer has not been demonstrated so far. Indeed there are probably two types of cancers that could derive from SSAs, one with good prognosis and one with bad prognosis (Phipps AI et al., Gastroenterology 2014).

The confusion that is present in this paragraph seems to derive from the unclear work hypothesis of this study, and it is reflected also in the unfocused Discussion.

Finally, they looked at the expression of the 5 most strongly induced and 5 most strongly reduced genes upon TGFbeta treatment in TA organoid cultures in their previously published transcriptomic data on 6 SSAs vs 7 TAs. Oddly, the 5 genes induced by TGFbeta were more expressed in SSAs than in TAs (and viceversa for the 5 reduced genes). This seems counterintuitive since both TA and BRAFV600E organoids had a robust EMT response. Evidently, these 10 genes are not related to the EMT response. If so, which type of response are they related to?

The insignificance of this finding is shown in the next step: the analysis of these genes' expression level in an independent series of SSAs and TAs. This was done only for 4 of the 10 genes, see Fig EV4C, and in two of them (50%) the difference btw SSAs and TAs was not significant.

In conclusion, the work hypothesis of this study was based on uncertain evidence from a previously published work, the experiments were done using a dubious model (BRAFV600E organoids), a likely-inappropriate mix of gene expression signatures from different studies and tumors (even from different organs) was applied, and validation experiments on tissue samples were not correctly performed. All this explains why Abstract and Discussion are unclear.

Referee #3 (Remarks):

The authors studied the effect of TGF-beta at the premalignant state of colorectal cancer development. They showed that after TGF-beta exposure RAS pathway wild-type tubular adenoma organoids upregulated BIM and underwent apoptosis, while RAS-mutated and normal organoids reduced proliferation without signs of apoptosis. Cells surviving TGF-beta treatment showed morphological and molecular features of EMT, including expression of genes related to the mesenchymal colon cancer subtype.

Sessile serrated adenoma seems to be the precursor lesion of the mesenchymal colon cancer subtype. Since the establishment of organoids from sessile serrated adenoma appears difficult, the authors produced by CRISP-Cas-9 technology organoids carrying mutated BRAF. Such BRAF-mutated organoids responded to TGF-beta by slowing down proliferation and acquiring EMT genes, suggesting that BRAF-mutant sessile serrated adenomas may exploit TGF-beta to increase EMT and aggressiveness. Finally, the authors investigated by gene expression profiling the correlation between TGF-beta-induced and adenoma genes. They first highlighted a correlation between the top 5 genes upregulated and downregulated by TGF-beta with those of sessile serrated and tubular adenomas, respectively. Then they generated a TGF-beta signature that was able to (i) segregate tubular from sessile serrated adenomas, (ii) cluster most of the mesenchymal colon cancer samples from those of the other subtypes, and (iii) predict recurrence-free survival in two independent datasets.

The manuscript is very interesting because it addresses some critical issues concerning the biology of colorectal cancer, bridging nicely TGF-beta and sessile serrated adenomas. However, there are some points that need clarification.

1. Since the response to TGF-beta seems to be dictated by the primary genetic lesion, so that KRAS- and BRAF-mutated samples do not undergo apoptosis, the reader may not get clear the difference between sessile serrated adenomas and RAS-mutated tubular adenomas in terms of TGF-beta response and cancerogenesis. The authors should better explain the genetic and epigenetic differences in this context.

2. The authors show that CDX2 is strongly downregulated by TGF-beta (Fig. 4). They should discuss their findings in the light of recent data indicating the existence of a small subset of CDX2-

negative stage II and III colorectal tumors with poor negative prognosis linked to the immature phenotype of cancer cells (Dalerba NEJM, 2016).

1st Revision - authors' response

24 March 2016

Point-by-point reply to the reviewer comments. Fessler *et al.* “TGFb induces a serrated-like phenotype in normal and adenomatous colon organoids”, MS#EMM-2016-06184.

We would like to take the opportunity to thank the expert reviewers and the editor for evaluating our manuscript and providing valuable criticism that has helped us to improve the quality of our work. We are confident that we have been able to address all points that have been raised by the reviewers. Please find below a point-by-point reply.

Referee #1:

This reviewer describes our work as “*logical and clear and a pleasure to read*” “*worthy work shedding light on diverse effects of TGFB signalling, as well as genesis of serrated colon tumors*” and states that “*the model is adequate though association to CMS4 stretched as mentioned in remarks*”

The only concern raised relates to the extrapolation to CMS4 as “*it is not really discussed that most BRAF mutations occur in CRCs other than type CMS4*” “*Thus, how well the two actually come together in real life?*”

We appreciate the time and effort of this reviewer to evaluate our work and the overall positive remarks on our manuscript.

We thank the reviewer for raising this critical point. We agree that based on molecular characteristics, such as *BRAF* mutations, some CMS4 tumors are predicted to arise from precursor lesions other than SSAs. Conversely, we also agree with the reviewer that not all *BRAF*^{V600E}-mutant SSAs will inevitably progress to CMS4 tumors, but that some of these precursor lesions will likely spawn CMS1 CRCs instead, as pointed out by reviewer #2 as well.

To address this point, we first sought to increase our sample size of precursor lesions. To this end, we isolated and characterized additional TAs (for a total of n = 15), as well as additional SSAs (for a total of n = 12), more than doubling the number of previously analyzed specimen. We generated gene expression profiles for these additional samples and classified these precursor lesions into the CMSs (new **Fig EV3B** in the manuscript). Previously, we had broadly stratified these lesions into either epithelial or mesenchymal. In the revised manuscript we decided to extend the analysis to include all consensus molecular subtypes (CMS1-4). As reported previously, almost all TAs classified as CMS2, with the exception of two TA samples that are related to CMS3.

Interestingly, due to the higher number of samples analyzed, the gene expression-based classification of the new panel of SSAs revealed that it in fact contains two subsets: while one subset is closely related to CMS4, as indicated in the original manuscript, another subset relates to CMS1 and is thus predicted to develop into this type of CRC instead. This gene expression-based classification therefore confirms the conclusion that *BRAF*^{V600E}-mutant SSAs can develop into either good or poor prognosis CRC that has been drawn previously using molecular characteristics (Phipps *et al.* Gastroenterology 2015; Jass, Histopathology 2007).

As in the previous version of the manuscript, we next used our TGFb signature and found that it effectively segregates TA from SSA lesions also in the now expanded set of precursor lesions (included as **Fig 4B** in the manuscript). Moreover, the majority of the top 10 TGFb regulated genes (up and down) is significantly differentially expressed between TAs and SSAs (**Fig 4A** in the manuscript), suggesting that TGFb signaling is indeed active in SSA lesions.

To confirm this gene expression-based analysis we evaluated the expression of the known TGFb target gene ZEB1 on protein level, which revealed that ZEB1 is indeed expressed in epithelial cells of SSA but not of TA precursor lesions (now included in the manuscript as **Fig EV3A**), pointing to an active TGFb signaling pathway in SSA precursor lesions.

Finally and most importantly, closer examination of the stratification with our TGFb signature indicates two separate clusters of SSA samples. These two subsets perfectly co-align with the SSA samples that are predicted to progress to either CMS1 (further referred to as CMS1-SSAs) or CMS4 CRC (further referred to as CMS4-SSAs; **Fig 4B**). Also at the single gene level, CMS4-SSAs express significantly higher levels of TGFb target genes compared to CMS1-SSAs (**Fig 4C**). Therefore, we believe these data indicate that SSAs can indeed progress to either CMS1 or CMS4 tumors and that high activity of the TGFb signaling pathway is decisive in directing these pre-neoplastic lesions to the mesenchymal CMS4 of CRC (**Fig 4D**). In our view, these new data significantly underline the importance of our proposed model in which TGFb plays a critical role in the development of CMS4, the most aggressive type of CRC.

Detailed:

Page 7 and 8 please provide information of the mutation status at organoid TAI earlier, the reader is a bit lost why this behaved differently and in the current version needs to wait for the explanation.

We apologize for structuring the text not optimally and have now rephrased this paragraph, also including a statement on the fact that we think this particular culture might represent a later stage in the classical path of CRC development compared to the *KRAS* wildtype TA organoid cultures.

Page 10 "genes present in a poor prognosis signature associated with TGFb signalling were highly enriched in samples belonging to the mesenchymal CMS and in SSA lesions" this sounds like circular reasoning.

We agree with the reviewer that this is partly circular reasoning when looking at the mesenchymal CMS4 CRC samples (i.e. these are associated with poor prognosis and display expression of TGFb signaling components, thus selecting TGFb-induced genes should result in poor prognosis again). However, this is not necessarily the case for the SSA samples where we observe a similar enrichment of this TGFb signature. Yet, we decided to leave out this figure in our revised manuscript as it is a re-iteration of what was already published and because reviewer #2 had strong objections regarding this figure. Moreover, the TGFb response in SSA samples is shown extensively in this study using different approaches.

Page 16 CRC tumors sounds odd; instead CRCs or colorectal tumors plz.

We have changed the text accordingly.

Referee #2:

This reviewer questions the validity of the organoid as model for the serrated pathway and states “An ideal, serrated adenoma organoid model would be that established directly from SSAs”.

Moreover, the role of TGFbeta in inducing a serrated-like phenotype is questioned and the title considered misleading.

We appreciate the time and effort of this reviewer to evaluate our work and value the useful comments on our data.

Indeed, our TA and *BRAF*^{V600E} organoid cultures do not adopt a serrated contour upon TGFb treatment. Since no literature is available on how organoid cultures of SSAs might look *in vitro*, it is difficult to judge whether this morphology is a crucial phenotype for SSA organoid cultures or may depend on supporting stromal cells or extracellular matrix not present in such cultures.

As the reviewer points out, the word “phenotype” in the title might be misleading and to better reflect the new data we have now changed the title to “TGFb signaling directs serrated adenomas to the mesenchymal colorectal cancer subtype”.

Indeed, the article we cite regarding the lack of SSA organoid cultures is a review article and thus does not feature experimental proof. We decided to cite this article, as it is a very recent review discussing this issue. Unfortunately, negative results such as the unsuccessful establishment of organoids cultures are infrequently reported and thus an experimental citation for this issue does not seem to be available.

While we agree with the reviewer (as we also describe in the discussion) that it will be important for the field to explore ways of establishing SSA organoid cultures, we respectfully disagree with the reviewer that *BRAF*^{V600E} organoid cultures are not a suitable model for SSA precursor lesions since they lack other mutations. It has been extensively reported that *BRAF*^{V600E} is the initiating mutation in SSAs (Leggett & Whitehall, Gastroenterology 2010; Snover, Human Pathology 2011) and conditional *Braf*^{V600E} mouse models have been demonstrated to recapitulate the serrated phenotype observed in patients (Carragher *et al.* EMBO Molecular Medicine 2010).

Indeed, additional genetic and most likely epigenetic alterations might be present in SSAs in patients, which have progressed further than the initial hit in the *BRAF* gene. However, we rather feel that the *BRAF*-mutant organoids are in fact the optimal model to study the effect of TGFb on the earliest stage of disease. This is similar to, for instance, the *Apc*^{min} mouse model, which is widely considered to be an appropriate model for early stages of classical CRC. Thus, while we agree that the *BRAF*-mutant organoids are not identical to SSAs (SSAs frequently display DNA hypermethylation, which is not present in our genetically engineered organoids as discussed on **page 17** of the manuscript), we rather consider this an advantage as it allows for the detection of early events in the serrated pathway and enables us to study cells that may be more naïve in the sense that they have not yet been exposed to environmental cues – such as TGFb. In the revised manuscript we have extended on the data generated with these organoids (Fig 5) (see below).

To confirm the validity of our approach, we also generated gene expression profiles of *BRAF*^{V600E}-mutant organoids with and without TGFb treatment. We show that the TGFb response is highly correlated to the TGFb response derived from TA organoids (included in the manuscript as **Fig EV5A**), indicating that TGFb induces a similar cellular program in our engineered model system as in “naturally arisen” precursor lesions. Moreover, genes upregulated upon TGFb in this system are more highly expressed in CMS4-SSAs compared to CMS1-SSAs (**Fig 5H** and **Fig EV5B** in the manuscript), and most importantly, this distinction also holds true for *BRAF*-mutated CMS1 and CMS4 CRC samples of the TCGA dataset (**Fig 5I** and **Fig EV5C** in the manuscript).

Integrating all of this data, we believe that our organoid cultures indeed represent a good model system for the following reasons:

1. Our data confirm that SSAs can progress to a good and a poor prognosis CRC subtype. This has previously been suggested based on molecular markers (Phipps *et al.* Gastroenterology 2015; Jass, Histopathology 2007). Making use of a more comprehensive gene expression-based classification strategy, we show that SSAs are indeed predicted to progress to either CMS1 or CMS4 tumors.
2. An *in vitro* TGFb response signature can separate SSA from TA lesions and the expression of individual genes suggests that TGFb signaling is more active in SSA compared to TA lesions. Interestingly, based on gene expression the TGFb signaling pathway is predicted to be more active in CMS1 and CMS4 tumors compared with CMS2 and CMS3 carcinoma samples (Guinney *et al.* Nature Medicine 2015).
3. The *in vitro* TGFb signature is also able to segregate CMS1-SSA from CMS4-SSA samples with the latter expressing higher levels of TGFb-induced genes, again paralleling the observation in CMS1 and CMS4 tumors: TGFb signaling activity is highest in CMS4 tumors, followed by a more moderate activity in CMS1 tumors, while the lowest activity is present in CMS2 and 3 (Guinney *et al.* Nature Medicine 2015).
4. The TGFb signatures derived from TA and *BRAF*^{V600E}-mutant organoids are highly correlated, highlighting the conformity of our engineered model systems with adenomas that formed in patients. Additionally, the gene expression differences are mirrored in CMS1-SSA/CMS1-CRC and CMS4-SSA/CMS4-CRC samples.

This reviewer is, like reviewers #1 and #3, also questioning the KRAS mutant adenomas. He/she states “*Therefore, it seems that the presence of such mutations -whether in KRAS or BRAF- changes the cellular response to TGFbeta, from an apoptotic to a growth arrest phenotype. However, KRAS mutations are rare in SSAs, therefore the authors cannot conclude that these phenotypes are typical of the serrated pathway.*”

We apologize for not having pointed out the meaning of the *KRAS*^{G12V} organoid culture more clearly. We do not think that this culture represents another form of SSA. The polyp used to isolate this culture was obtained from a FAP patient and the organoids grow without addition of stimuli of the WNT pathway such as WNT3A and R-Spondin-1 conditioned medium. We therefore conclude that this culture is representative of a TA. However, it contains a *KRAS* mutation, which is commonly believed to be a hit during progression from early stage adenoma to a more advanced neoplastic lesion and known to protect against TGFb-induced apoptosis. We therefore think that this particular culture represents a late stage adenoma or very early stage carcinoma. The observation that, unlike all other TA cultures, this particular organoid culture does not undergo cell death upon TGFb treatment suggests that TGFb signaling is only detrimental for TAs at early stages, but once progressed to a more advanced lesion the apoptotic effect is abrogated. We have now rephrased this paragraph on **page 7** to convey this message more clearly.

Next to the questions on the model itself this reviewer, like reviewer #1, doubts the validity of using gene-expression analyses to predict the development of adenomas.

“finding some overlap among transcriptomic signatures does not justify making a direct pathogenetic link between different types of tumors”

For an answer to this crucial question we would like to point this reviewer to the answers given above (pages 1 and 2 of this document) and the novel data included in the manuscript. Answers to the more specific points are given below.

This reviewer states “*Although I cannot access the accuracy of the enrichment analysis performed in this study, it is a dangerous game to make conclusions by comparing gene expression data from several different studies, even one from ovarian cancers. Even more dangerous is then to correlate these findings with what is generally known on the prognosis of such tumors. This problem emerges clearly from sentences like these:*

page 11 and 12: These data indicate that precursor lesions from distinct backgrounds respond differentially to TGFbeta; stimulation early in tumor development. The apoptotic response dominated in classical TAs, yet, in both TA and BRAFV600E-mutated organoids a robust EMT response was observed upon TGFbeta; treatment. In the case of TA organoids, this seems to be a selection-based response, whereas the majority of BRAFV600E-mutated organoids is able to induce the EMT process. Thus, BRAFV600E-mutant SSAs might benefit from TGFbeta; pathway activation directing these precursor lesions to a more aggressive phenotype, marked by the induction of the EMT program, toward the mesenchymal, poor prognosis CRC subtype.

First sentence: The authors have not investigated precursor lesions from different background; they have studied organoids in vitro.

We regret that this part of the text was perceived in such a negative way by the reviewer.

The differences in background refer to the genetic makeup of the distinct organoid cultures, and we have rephrased this sentence on **page 14** accordingly. Yet, we respectfully disagree that we have only looked at organoids *in vitro*. We have studied gene and protein expression of TAs and SSAs directly from patients. In the revised version this aspect is expanded significantly and now represents to the best of our knowledge the largest set of such adenomas reported in literature so far. These adenomas are used to validate the findings on organoids and strengthen the concept that TGFb is an important player in the serrated pathway inducing mesenchymal features.

Using gene set enrichment analyses based on published signatures as well as KEGG pathway analyses is commonly performed when analyzing transcription data. GSEA considers a wide range of genes and it has therefore been shown to reliably transfer to other tissues in most cases. Yet, we agree with the reviewer, that the fact that these KEGG pathways and signatures are frequently derived from distinct tissues warrants caution in interpreting such data. Since this particular GSEA does in fact not add to what is already described in literature for the TGFb target genes in CMS4 and SSA tumor samples (Guinney *et al.* Nature Medicine 2015; De Sousa E Melo *et al.* Nature Medicine 2013), we omitted these graphs in the revised version of the manuscript. More importantly, our current data now provide the optimal TGFb signatures derived in the tissue of interest.

Second and third sentences: they first write that both TA and BRAFV600E organoids had a robust EMT response (second sentence), but conclude the third sentence with "whereas the majority of BRAFV600E-mutated organoids is able to induce the EMT process". Also, what is a "selection-based response"?

We still believe that this part is very important as it highlights the differences between selection and induction. A selection-based response refers to the fact that most TA organoids die upon TGFb treatment, however, a small proportion of cells which is able to survive the TGFb stimulation and undergo EMT is selected for. In contrast, all cells in $BRAF^{V600E}$ -mutant organoids seem to be able to induce an EMT process. We have changed the text to convey this – in our eyes important – message more clearly (see **page 14** of the manuscript). To highlight this difference in apoptotic response versus EMT induction, we now included data showing the level of apoptosis in the control and TGFb-treated condition of the organoid cultures. The vast majority of cells in the TA organoid cultures (without mutant *KRAS*) undergo apoptosis, suggesting that only the cells that escape death will undergo the EMT process. In contrast, in the $BRAF^{V600E}$ -mutant organoids, the level of apoptosis is not increased in the TGFb-treated compared with the control condition, thus EMT is the main response to treatment (now included in the manuscript as **Fig EV4**).

Forth sentence: why would the SSAs benefit from the TGFbeta pathway activation? because they would stop growing instead of dying via apoptosis? Also, the direct relationship between SSAs and the poor prognosis, mesenchymal colon cancer has not been demonstrated so far. Indeed there are probably two types of cancers that could derive from SSAs, one with good prognosis and one with bad prognosis (Phipps AI et al., Gastroenterology 2014).

The confusion that is present in this paragraph seems to derive from the unclear work hypothesis of this study, and it is reflected also in the unfocused Discussion.

We indeed think that growth arrest would be more beneficial to premalignant lesions than the finality of death by apoptosis. Firstly, growth arrested cells could persist for extended periods of time and resume growth upon appropriate stimulation. Secondly, as we state in the discussion, the *BRAF*^{V600E}-mutant organoid culture does not display methylation at the *CDKN2A* locus, which is often observed in SSA lesions derived from patients. As we discuss on **page 17**, the common inactivation of this gene *in vivo* is likely to allow the cells to circumvent potential growth arrest phenotypes induced by TGFb while exploiting the pro-malignant effects of TGFb signaling and progress to a malignant stage.

We agree with the reviewer that SSAs progressing only to CMS4 tumors is an oversimplification and as described on pages 1 and 2 of this rebuttal document, we have now expanded the set of TA and SSA samples and indeed confirm molecular marker-based studies with our gene expression-based analysis. SSAs are predicted to progress to CMS1 or CMS4 tumors, and most importantly, our data suggest that high activity of the TGFb signaling pathway directs these precursor lesions to the CMS4 of CRC. For a detailed description of the new data, please see the response to reviewer #1 on pages 1 and 2 of this document.

Initially, we were eager to discuss our findings in light of different concepts and studies on the most important topics of this manuscript. We apologize that this led to the perception that the discussion is unfocused and have now changed it to be more purposeful.

Finally, they looked at the expression of the 5 most strongly induced and 5 most strongly reduced genes upon TGFbeta treatment in TA organoid cultures in their previously published transcriptomic data on 6 SSAs vs 7 TAs. Oddly, the 5 genes induced by TGFbeta were more expressed in SSAs than in TAs (and viceversa for the 5 reduced genes). This seems counterintuitive since both TA and BRAFV600E organoids had a robust EMT response. Evidently, these 10 genes are not related to the EMT response. If so, which type of response are they related to?

We apologize for not conveying the point of this figure clearly enough. The 10 genes we looked at are regulated by TGFb signaling, yet, they were not selected for their association with any biological program, but purely based on their ranking in the TGFb response signature (five most strongly induced and five most strongly reduced genes). Thus, they can be – as in the case of FN1 – but do not necessarily have to be related to the EMT program, as TGFb is known to stimulate other programs next to EMT.

Indeed, both TA and *BRAF*^{V600E}-mutant organoids are able to induce an EMT response, however, in this figure we analyzed the expression of these genes in samples from patients. We assume that TGFb is not present in *in vivo* TAs, otherwise they would most likely undergo apoptosis. By contrast, our data suggest that TGFb signaling is active in SSA lesions. Therefore, the expression of the TGFb target genes in TAs represents a baseline in a non-stimulated setting. In SSAs, where TGFb signaling is presumably active, the target genes are regulated just as in cultures stimulated with TGFb: those genes induced upon TGFb treatment are more highly expressed in SSAs compared to TAs, whereas those downregulated upon TGFb stimulation are also downregulated in the *in vivo* situation in SSAs compared to TAs. Intriguingly, this appears even stronger in the SSAs that are predicted to be CMS4-like.

Finally this reviewer considers our validation by qPCR to reveal insignificance. *“The insignificance of this finding is shown in the next step: the analysis of these genes' expression level in an independent series of SSAs and TAs. This was done only for 4 of the 10 genes, see Fig EV4C, and in two of them (50%) the difference btw SSAs and TAs was not significant.”*

Due to the additional samples of TAs and SSAs of which we generated gene expression profiles, we were now able to expand this analysis to all the genes for a larger amount of samples. Almost all genes display a significantly differential expression between TA and SSA samples. Importantly, FN1 – whose expression is not significantly different between SSA and TA samples – becomes highly significant when SSA samples are separated into CMS1-SSAs and CMS4-SSAs.

The overall conclusion of this reviewer is rather harsh, questioning the previously published work, the hypothesis addressed in this study using a *“dubious model”*, the improper execution of validation experiments and the use of gene expression signatures. *“All this explains why Abstract and Discussion are unclear.”*

Although we anticipate that we will not be able to convince this reviewer of our model, we hope that the revised manuscript conveys our message more clearly. In addition, we hope that this reviewer recognizes that we have in fact generated a unique set of gene expression data from adenomas directly from patients and this forms the basis of the *in vitro* work and validates our claims. It will be extremely difficult to prove that a SSA will eventually end up in a CMS4-like tumor when exposed to TGF β , as such a longitudinal follow up is obviously not feasible in patients. However, the classical studies by Fearon and Vogelstein used a similar approach focused on just a few mutations and a large series of adenomas and carcinomas from multiple patients to derive what is now broadly accepted as the classical pathway to CRC. We believe our experiments, combining *in vivo* observations with *in vitro* studies, add substantially to our understanding of the serrated pathway and we hope that the reviewer will be convinced that the revised version of the manuscript with the data extension provides important insight into our knowledge of this pathway.

Referee #3 (Remarks):

This reviewer views our manuscript as *“very interesting because it addresses some critical issues concerning the biology of colorectal cancer, bridging nicely TGF-beta and sessile serrated adenomas.”*

We thank the reviewer for evaluating our study and the appreciation of our work as well as the insightful summary.

However, there are some points that need clarification.

The first point raised is on the distinction between KRAS mutant TAs and BRAF-mutant SSAs and the response to TGF-beta *“The authors should better explain the genetic and epigenetic differences in this context.”*

We thank the reviewer for this suggestion. We believe that the KRAS^{G12V}-mutated TA1 represents a culture of a more advanced adenoma or early carcinoma, but is not related to SSAs. We rephrased the text on **page 7** of the manuscript to convey the message more clearly. Please also see the reply to reviewer #2 in the bottom paragraph on page 4 of this document.

The second point concerns the downregulation of CDX2 by TGF-beta (Fig 3) and how this relates to the recent publication of Dalerba *et al.* (NEJM, 2016).

We thank the reviewer for this interesting question. This study provides intriguing insight into the makeup of poor prognosis CRCs and indeed our findings should be discussed in the light of these results. Low/absent expression of CDX2 has previously been described to identify tumors of the mesenchymal subtype (De Sousa E Melo *et al.* Nature Medicine 2015). As this subtype relates to poor prognosis also in datasets comprising stage II patients only, this is highly concordant with the above-mentioned study. However, it is not clear how and at what stage in tumor progression CDX2 expression is lost. Our data suggest that TGFb signaling downregulates CDX2 expression, which could be installed early in tumor progression. Furthermore, the induction of an EMT phenotype, which has also been linked to the stem-like and thus undifferentiated state, can be stimulated by TGFb and could therefore explain both low levels of CDX2 expression and the immature phenotype of cancer cells. This has now been included in the discussion on **page 17**.

2nd Editorial Decision

13 April 2016

Thank you for the submission of your revised manuscript to EMBO Molecular Medicine.

We have now received the enclosed reports from the referees that were asked to re-assess it. As you will see the reviewers are now globally supportive and I am pleased to inform you that we will be able to accept your manuscript pending the following final amendments:

- 1) As you will see, Reviewer 2, who was initially quite reserved, is now positive. S/he does, however, have a few remaining requests for your action, mostly aimed at clarifying and to provide more experimental details, and for which I will not requiring further experimentation. I would like you to carefully consider, and respond to each point, introducing the appropriate textual changes in the manuscript where necessary. Provided you respond to each point, I am prepared to make an editorial decision on your next, final version of your manuscript. To this effect, please remove the red lettering from your current version (no longer needed) and highlight the new changes.
- 2) Every published paper now includes a 'Synopsis' to further enhance discoverability. Synopses are displayed on the journal webpage and are freely accessible to all readers. They include a short standfirst as well as 2-5 one sentence bullet points that summarise the paper. Please provide the synopsis including the short list of bullet points that summarise the key NEW findings. The bullet points should be designed to be complementary to the abstract - i.e. not repeat the same text. We encourage inclusion of key acronyms and quantitative information. Please use the passive voice. Please attach this information in a separate file or send them by email, we will incorporate it accordingly. You are also welcome to suggest a striking image or visual abstract to illustrate your article. If you do please provide a jpeg file 550 px-wide x 400-px high.
- 3) We are now encouraging the publication of source data, particularly for electrophoretic gels and blots, with the aim of making primary data more accessible and transparent to the reader. Would you be willing to provide a PDF file per figure that contains the original, uncropped and unprocessed scans of all or at least the key gels used in the manuscript? The PDF files should be labeled with the appropriate figure/panel number, and should have molecular weight markers; further annotation may be useful but is not essential. The PDF files will be published online with the article as supplementary "Source Data" files. If you have any questions regarding this just contact me.

Please submit your revised manuscript within two weeks. I look forward to seeing a revised form of your manuscript as soon as possible.

***** Reviewer's comments *****

Referee #1 (Remarks):

The revised version has sufficiently addressed the issues raised by this reviewer.

Referee #2 (Remarks):

This revision is more clear than the previous version. Additional data have improved the message.

Comments:

1) First sentences in Abstract and in "The paper explained" should be corrected: "The heterogeneous nature of CRC severely complicates patient stratification..." It is in fact the heterogeneous nature of a disease that allows patient stratification.

2) Page 7: From how many FAP patients were established TA1 to TA5?

3) Page 7: "...to select for transformed cells possessing an activating mutation in the WNT pathway" Do they mean an APC mutation in homozygous configuration? Please, make this point more precise. Also, specify the origin of the normal mucosa used to establish organoids (healthy individuals?).

4) All organoid tissues (normal mucosa, TA with or without KRAS mut, the BRAF mut organoid) showed a similar TGFbeta response. One would have expected a certain level of TGFbeta signature in the BRAF mut organoid even in the absence of TGFbeta treatment (if this would be a suitable model of SSA). Please, comment on this point, a new figure with a clustering analysis of the transcriptomic data from all the organoids before and after exposure to TGFbeta could help.

(Maybe, SSA organoids might be established with a certain level of TGFbeta in the culture medium ??)

5) Whenever the BRAF mutated organoid culture is mentioned in the manuscript, it should not be called "organoids" since this way seems that different independent organoid cultures of this type were established and analyzed. Actually there are no replicates.

6) page 16: "We show that the genetic background of preneoplastic lesions dictates the response to this signaling molecule and that TGFbeta influences subtype affiliation" If the authors refer to the organoids used in this study, all of them showed a similar response to TGFbeta. Therefore, this paragraph has to be revised (see also comment no. 4).

7) If available, for the TAs and SSAs of Figure 4B, the KRAS and BRAF status and some clinical information should be provided. Also indicate if the lesions were from different (27) patients.

8) In figure 3E, the enrichment score does not seem very high and the curve goes below 0 on the right side. Is this considered really significant? How many SSAs were used for this analysis? Should this also be split in two curves, one for the CMS4-SSA and one for the CMS1-SSAs?

Referee #3 (Comments on Novelty/Model System):

This is a nice effort. Human early cancers are difficult to study and poorly investigated.

Referee #3 (Remarks):

The authors have improved the manuscript by reasonably addressing all the points raised to the previous version.

2nd Revision - authors' response

19 April 2016

Point-by-point reply to the reviewer comments. Fessler *et al.* "TGFb signaling directs serrated adenomas to the mesenchymal colorectal cancer subtype", MS#EMM-2016-06184-V2.

We would like to thank the expert reviewers and the editor for re-evaluating our manuscript and the positive assessment of our study. We are confident that we have been able to address the remaining concerns of reviewer #2 by improving the clarity of the text and providing additional information regarding the adenoma samples used in this study. Please find below a point-by-point reply.

Referee #1:

The revised version has sufficiently addressed the issues raised by this reviewer.

We again wish to thank this reviewer for his/her constructive comments.

Referee #2:

This revision is more clear than the previous version. Additional data have improved the message.

We thank this reviewer for his/her time and effort to re-evaluate our work and for the additional comments to improve the clarity of our manuscript.

Comments:

1) First sentences in Abstract and in "The paper explained" should be corrected: "The heterogeneous nature of CRC severely complicates patient stratification..." It is in fact the heterogeneous nature of a disease that allows patient stratification.

We appreciate this valuable comment and have changed the text accordingly.

2) Page 7: From how many FAP patients were established TA1 to TA5?

The 5 TA organoid cultures were established from 4 different patients. TA1 and TA2 were derived from independent polyps of the same patient, and TA3-TA5 from 3 additional patients. This information is now included in the materials and methods section "Organoid isolation, culture, and treatment" on pages 21 and 22 of the manuscript.

3) Page 7: "...to select for transformed cells possessing an activating mutation in the WNT pathway" Do they mean an APC mutation in homozygous configuration? Please, make this point more

precise. Also, specify the origin of the normal mucosa used to establish organoids (healthy individuals?).

The TA organoid cultures grow without the need for WNT pathway activation with WNT3A and R-Spondin-1, which points to the constitutive activation of the WNT pathway in these cultures. We have not determined the *APC* or *CTNGB1* mutation status for the distinct organoid cultures and constitutive activation of the WNT pathway can also be achieved non-genetically for instance by epigenetic modifications. To better reflect the different possibilities of WNT pathway activation, we have now changed this sentence to "...to select for transformed cells in which the WNT pathway is constitutively active."

Normal mucosa to establish organoids has been obtained from resection material from patients with CRC. The normal tissue was collected from a part of the mucosa ≥ 10 cm apart from the cancer tissue. This information is now included in the materials and methods section "Organoid isolation, culture, and treatment" on page 22 of the manuscript.

4) All organoid tissues (normal mucosa, TA with or without KRAS mut, the BRAF mut organoid) showed a similar TGFbeta response. One would have expected a certain level of TGFbeta signature in the BRAF mut organoid even in the absence of TGFbeta treatment (if this would be a suitable model of SSA). Please, comment on this point, a new figure with a clustering analysis of the transcriptomic data from all the organoids before and after exposure to TGFbeta could help.

(Maybe, SSA organoids might be established with a certain level of TGFbeta in the culture medium ??)

We respectfully disagree with the reviewer that the *BRAF*^{V600E}-mutant organoid culture should display TGFb signaling activity even in the absence of TGFb treatment. The *BRAF*^{V600E}-mutant organoid culture is not identical to SSAs, which may contain additional genetic and/or epigenetic alterations next to the *BRAF*^{V600E} mutation. We believe and describe that it rather represents a model for the earliest stage of the serrated neoplasia pathway, thus being more naïve in the sense that it has not yet been exposed to environmental cues – such as TGFb. Our model suggests that the *BRAF*^{V600E}-mutant organoid culture, when exposed to TGFb, displays a gene expression profile that is similar to SSAs and can deal with the apoptosis-inducing capacity of TGFb. We have clarified this on page 14 stating "We therefore conclude that *BRAF*^{V600E} mutations in combination with a microenvironmental TGFb signal could be the underlying pathway towards the induction of SSAs."

Of note, we have tried to establish SSA cultures in the presence of low and high levels of TGFb, but unfortunately have not succeeded to derive such cultures, yet.

5) Whenever the BRAF mutated organoid culture is mentioned in the manuscript, it should not be called "organoids" since this way seems that different independent organoid cultures of this type were established and analyzed. Actually there are no replicates.

We thank the reviewer for pointing this out and have changed the phrase "*BRAF*^{V600E}-mutant organoids" to "*BRAF*^{V600E}-mutant organoid culture" throughout the text.

6) page 16: "We show that the genetic background of preneoplastic lesions dictates the response to this signaling molecule and that TGFbeta influences subtype affiliation" If the authors refer to the organoids used in this study, all of them showed a similar response to TGFbeta. Therefore, this paragraph has to be revised (see also comment no. 4).

We agree with the reviewer that the organoids used in this study show a similar response with regard to EMT, yet, the apoptotic response is markedly different. TAs respond to TGFb with EMT

induction, but, as we show in Figure EV4, the vast majority of cells undergoes apoptosis. Importantly, apoptosis is no longer induced if an activating mutation in *KRAS* is present in TA organoids as well as in an organoid culture carrying the *BRAF*^{V600E} mutation. Therefore, we believe that the genetic background is indeed responsible for dictating the response to TGFb. To convey more clearly that the phenotype induced by TGFb partially overlaps we have rephrased the above sentence to “We show that the genetic background of pre-neoplastic lesions dictates the dominating response to this signaling molecule, changing it from a largely apoptotic response in WNT pathway-activated TAs to a dominant EMT response in mitogen-activated protein kinase (MAPK) pathway mutant cells. In addition, our data shows that TGFb influences subtype affiliation.”.

7) *If available, for the TAs and SSAs of Figure 4B, the KRAS and BRAF status and some clinical information should be provided. Also indicate if the lesions were from different (27) patients.*

We agree with this reviewer that this is important information and have now included the mutation status of the *KRAS* and *BRAF* oncogenes as well as the CIMP status of the polyps as **Figure EV3C**. We have added information regarding the generation of this data in the Appendix materials and methods section “CIMP, *BRAF*, and *KRAS* mutation status analysis in patient-derived adenoma samples” on pages 4 and 5 of the appendix document. The 27 polyps were obtained from a total of 14 patients, which is now stated in the materials and methods section “Processing of TA and SSA samples” on page 21 of the manuscript.

8) *In figure 3E, the enrichment score does not seem very high and the curve goes below 0 on the right side. Is this considered really significant? How many SSAs were used for this analysis? Should this also be split in two curves, one for the CMS4-SSA and one for the CMS1-SSAs?*

The *P*-value for this GSEA is shown in the figure and is indeed significant. The fact that the line goes below the axis is not a sign of insignificance, but shows that not all genes of this gene set are upregulated following TGFb treatment of organoid cultures. The enrichment score is indeed lower as compared to for instance the one in Figure 3D, but the significance of enrichment is in part determined by the number of genes in the gene list as well as the number of samples that are compared and, as said, is still significant.

The second question suggests that the comparison may not be completely clear. We have used a gene set (500 most upregulated genes in SSA polyps compared to TAs) that was derived from 6 SSA samples (in comparison with 7 TAs) published previously in De Sousa E Melo *et al.* Nature Medicine 2013, GSE45270, which is stated in the materials and methods paragraph “Gene set enrichment analysis” on page 29 of the manuscript. Figure 3E shows that the genes identified to be expressed higher in SSAs have a strong similarity to the genes induced by TGFb in the organoid cultures. The CMS1-SSA vs CMS4-SSA comparison the reviewer is referring to is not really relevant at this point of the manuscript as this difference has not been introduced at this point, yet, but is in fact shown in a slightly different way in figure 5H where we now compare CMS1-SSAs with CMS4-SSAs with respect to up and downregulated genes in the TGFb-treated *BRAF*^{V600E}-mutant organoid culture. We have clarified this more extensively in the text on pages 10 and 11.

Referee #3:

Comments on Novelty/Model System:

This is a nice effort. Human early cancers are difficult to study and poorly investigated.

Remarks:

The authors have improved the manuscript by reasonably addressing all the points raised to the previous version.

We thank the reviewer for his/her appreciation of our manuscript and the improvements we made following his/her previous suggestions.

Corresponding Author Name: Jan Paul Medema

Manuscript Number: EMM-2016-06184